# Constitutive STAT5 activation regulates Paneth and Paneth-like cells to control *Clostridium difficile* colitis

Ruixue Liu[2,*], Richard Moriggl[3,4,5,*], Dongsheng Zhang[1,*], Haifeng Li[2], Rebekah Karns[1], Hai-Bin Ruan[6], Haitao Niu[2], Christopher Mayhew[7], Carey Watson[8], Hansraj Bangar[9], Sang-wook Cha[7], David Haslam[9], Tongli Zhang[12], Shila Gilbert[1], Na Li[2], Michael Helmrath[8], James Wells[7,10,11], Lee Denson[1,13], Xiaonan Han[1,2,13]

***Clostridium difficile*** **impairs Paneth cells, driving intestinal inflammation that exaggerates colitis. Besides secreting bactericidal products to restrain** ***C. difficile,*** **Paneth cells act as guardians that constitute a niche for intestinal epithelial stem cell (IESC) regeneration. However, how IESCs are sustained to specify Paneth-like cells as their niche remains unclear. Cytokine-JAK-STATs are required for IESC regeneration. We investigated how constitutive STAT5 activation (Ca-pYSTAT5) restricts IESC differentiation towards niche cells to restrain** ***C. difficile*** **infection. We generated inducible transgenic mice and organoids to determine the effects of Ca-pYSTAT5-induced IESC lineages on** ***C. difficile*** **colitis. We found that STAT5 absence reduced Paneth cells and predisposed mice to** ***C. difficile*** **ileocolitis. In contrast, Ca-pYSTAT5 enhanced Paneth cell lineage tracing and restricted Lgr5 IESC differentiation towards pYSTAT5+Lgr5−CD24+Lyso+ or cKit+ niche cells, which imprinted Lgr5hiKi67+ IESCs. Mechanistically, pYSTAT5 activated Wnt/β-catenin signaling to determine Paneth cell fate. In conclusion, Ca-pYSTAT5 gradients control niche differentiation. Lack of pYSTAT5 reduces the niche cells to sustain IESC regeneration and induces** ***C. difficile*** **ileocolitis. STAT5 may be a transcription factor that regulates Paneth cells to maintain niche regeneration.**

## Introduction

The prevalence of *Clostridium difficile* infection has increased in patients with inflammatory bowel diseases (IBDs) and has become a major healthcare burden over the past decade (Kaplan, 2015; Rao & Higgins, 2016). *C. difficile* infection is associated with increased disease severity and need for ileostomy or colectomy in patients with IBD (Chen et al, 2017); yet, preventive and therapeutic approaches are extremely limited by a lack of understanding of the essential cell types and key signaling proteins that are usurped in *C. difficile* infection to impair mucosal healing in IBD (Monaghan et al, 2015). Therefore, studying *C. difficile* infection in the context of IBD will directly impact the quest to treat and cure IBD.

*C. difficile* infection causes a persistent accumulation of enteric toxin A or cytotoxic toxin B and associated pro-inflammatory cytokines detained within intestinal mucosa, likely resulting in intestinal epithelial stem cell (IESC) niche degeneration and suppression of IESC regeneration (Farin et al, 2014; Leslie et al, 2015). The injured IESCs result in impaired intestinal epithelial (IEC) repair and reduced anti-microbial peptide production, which in turn drives persistent infection and mucosal inflammation progression to ileitis and/or colitis (Monaghan et al, 2015). STAT5-dependent JAK2 signaling is required for anti-inflammatory cytokine production and IEC repair, and mutations or single nucleotide polymorphisms in JAK2-STAT5 increase susceptibility to colitis and ileal Crohn's disease (Gilbert et al, 2012a; Huang et al, 2015; Chuang et al, 2016). *C. difficile* toxin has been implicated in suppression of the Wnt and JAK2-STAT5 pathways to impair IECs (Nam et al, 2012; Chen et al, 2018), but mechanistic studies are lacking.

Niche cells are located at the crypt bases that directly surround IESCs and provide a microenvironment that maintains Lgr5 IESC self-renewal (Sato et al, 2011; Rothenberg et al, 2012; Watt & Huck, 2013; Sasaki et al, 2016). IESCs and progenitor cells along with their regulatory secretory niche cells are thought to regulate crypt immune specialization to restrain infection and control the IEC healing response to inflammation (Barker, 2014; Mowat & Agace, 2014). In addition to secreting anti-microbial peptides for gut innate

[1]Division of Gastroenterology, Hepatology and Nutrition, Cincinnati Children's Hospital Medical Center (CCHMC), Cincinnati, OH, USA  [2]Key Laboratory of Human Disease Comparative Medicine, the Ministry of Health, Institute of Laboratory Animal Sciences, Chinese Academy Institute of Medical Sciences and Peking Union Medical College, Beijing, P.R. China  [3]Ludwig Boltzmann Institute for Cancer Research, Vienna, Austria  [4]Institute of Animal Breeding and Genetics, University of Veterinary Medicine, Vienna, Austria  [5]Medical University of Vienna, Vienna, Austria  [6]Department of Integrative Biology and Physiology, University of Minnesota Medical School, Minneapolis, MI, USA  [7]Division of Developmental Biology, CCHMC, Cincinnati, OH, USA  [8]Division of Pediatric Surgery, CCHMC, Cincinnati, OH, USA  [9]Division of Infectious Diseases, CCHMC, Cincinnati, OH, USA  [10]Division of Endocrinology, CCHMC, Cincinnati, OH, USA  [11]Center for Stem Cell and Organoid Medicine, CCHMC, Cincinnati, OH, USA  [12]Department of Pharmacology & Systems Physiology, University of Cincinnati College of Medicine, OH, USA  [13]Department of Pediatrics, University of Cincinnati College of Medicine, OH, USA

Correspondence: xiaonan.han@cchmc.org
*Ruixue Liu, Richard Moriggl, and Dongsheng Zhang contributed equally to this work

immunity to mediate the interaction with microbiota, Paneth cells act as defined niche cells of IESCs. In contrast, dysfunctional Paneth cells can serve as the site of origin for intestinal inflammation (Adolph et al, 2013). These reports indicate that specific Paneth cell phenotypes occur in intestinal diseases, such as colitis or enteric infection, and these phenotypes indicate either host prevention of intestinal injuries or exaggerated mucosal inflammation (VanDussen et al, 2014). Perhaps more intriguing, the misallocation of Paneth cells can be induced by various mucosal injuries or wound-healing factors (Nakanishi et al, 2016), suggesting that the sublineages of Paneth cells or Paneth-like cells may be differentiated to maintain the feed-forward loop of IESC regeneration (Schewe et al, 2016).

The biological efficacy of cytokines is often dependent on their ability to generate a sustained, rather than transient, stimulation of their target cells (Stark & Darnell, 2012). Persistent phosphorylated STAT5 (pYSTAT5) results in the maturation of mammary gland that maintains secretion (Xu et al, 2009). In our previous work (Gilbert et al, 2015), we found that Stat5 variants (STAT5a-ER) could be activated by different doses of tamoxifen (Tam) or by STAT5-activating cytokines or growth hormones (granulocyte-macrophage colony-stimulating factor [GM-CSF], c-Kit ligands, Leptin, Prolactin [Prl], and Growth Hormone [GH]), to represent physiological activation of cellular STAT5 (Grebien et al, 2008). In contrast, inducible constitutively active Stat5 (icS5) variants are "superactivatable" and can mimic persistent tyrosine kinase signaling independent of cytokine stimulation (Moriggl et al, 2005). Importantly, icS5 dosage can be controlled by chemical induction (Grebien et al, 2008; Gilbert et al, 2015). Low to intermediate levels of STAT5 activity confer self-renewal capacity to IESCs and hematopoietic stem cells (Wierenga et al, 2008; Gilbert et al, 2015), while higher or sustained STAT5 activation leads to progressive lineage differentiation and functional maturation, as in the case of mammary cell secretion (Yoo et al, 2015). Therefore, in vivo and in vitro icS5 are important tools for studying the effects of niche cytokine signaling upon IESC-dependent regeneration repair, as little is known about the mechanism of cytokine signaling that enables IESCs to restrict cell fates for specializing crypt immunity, such as architecture, anti-microbial secretion, and crypt cell hierarchy. Here, by differentiating human inducible pluripotent stem cells (iPSCs) or murine adult IESCs and utilizing compound mutant mice (Gilbert et al, 2012a, 2015), we defined a population of IESCs by which constitutive STAT5 activation (Ca-pYSTAT5) regenerated the niche, and unveiled the effects of defective JAK2-STAT5 signaling upon IESC niche cells, leading to susceptibility to enteric infection and ileocolitis.

# Results

### Lack of pYSTAT5 predisposes mice to *C. difficile* infection-induced ileitis and colitis

*C. difficile* causes diarrhea and pseudomembranous colitis via toxin A and cytokine-induced IEC integrity disruption (Johal et al, 2004), and potent toxin B cytotoxicity (Giesemann et al, 2008). Interestingly, apart from toxin-induced cytotoxicity, *C. difficile*

infection leads to the activation of a cascade of mucosal pro-inflammatory cytokines, including IFN$\gamma$, TNF$\alpha$, and IL1$\beta$, that result in recurrent colitis or ileitis (Abt et al, 2016). Some studies found that Paneth cell Defensins ($\alpha$-Defensins) inhibited toxin B possibly through promoting the unfolding of toxin protein, rendering it susceptible to proteolysis. This observation suggests a direct bactericidal role of Paneth cells in controlling hypervirulent *C. difficile* (Giesemann et al, 2008; Hing et al, 2013). Importantly, Paneth cells maintain host–microbiota homeostasis (Zhang et al, 2015a), as genetic defects in Paneth cells cause intestinal dysbiosis and increase susceptibility to both IBD (Wehkamp et al, 2007) and *C. difficile* ileitis or colitis (ileocolitis) (Giesemann et al, 2008). Together, these reports suggest a crucial role for host factors, such as IESC regeneration, Paneth cell differentiation, or crypt immune specification in the *C. difficile*-associated IBD severity.

Generally, *C. difficile* infection leads to scant exudate, surface colonization by *C. difficile*, submucosal neutrophil infiltrate, submucosal edema, IEC necrosis, goblet cell depletion, and transmural necrosis in the cecum and colon (Hing et al, 2013). We observed that *C. difficile*-infected IEC Stat5-deficient mice (VilCreER;Stat5$^{f/f}$; STAT5$^{\Delta IEC-/-}$) showed more severe cecal inflammation and colitis, characterized by more colonic goblet cell depletion, more severe pseudomembranous colitis and crypt necrosis, and worse mucosal coagulative necrosis than these lesions in *C. difficile*-infected controls (STAT5$^{+/+}$). In contrast, inducible activation of STAT5 in IECs (VilCreER;icS5; STAT5$^{\Delta IEC+++}$) alleviated cecal mucosal inflammation (Fig 1A and B). Additionally, significant weight loss (Fig S1A) and increased mortality was found in Stat5-deficient mice (Fig 1C). Importantly, we found that compared with controls (*C. difficile*-treated Lgr5CreER mice), *C. difficile*-infected Stat5-deficient mice (Lgr5CreER;VilCreER;Stat5) displayed virtually no Lgr5-GFP cell proliferation as measured by Lgr5$^+$ and Ki67$^+$ IECs at crypt bases whereas activated STAT5 (Lgr5CreER;VilCreER;icS5) induced remarkable Lgr5 IESC proliferation indicated by Lgr5$^+$ and Ki67$^+$ crypt IECs during *C. difficile* infection (Fig 1D and Fig S1B). Intriguingly, we found that STAT5-deficient mice are more susceptible to *C. difficile*-induced terminal ileitis (Fig 1E). *C. difficile*-induced ileitis in Stat5-deficient mice was characterized by ileal Paneth cell depletion, pseudomembrane formation, crypt abscess, and ileitis, while *C. difficile*-infected control mice did not show any enteritis (Fig 1F). In contrast, the activation of STAT5 in IECs robustly increased Paneth cell expansion (Fig 1E and F). These data indicate that the lack of pYSTAT5 induces *C. difficile* ileocolitis. Active STAT5 counteracts *C. difficile*-induced IEC or IESC injuries, possibly through promotion of niche-dependent Lgr5 IESC regeneration. Dextran sodium sulfate (DSS) induces colitis that may lead to colonic IEC damage and IESC regeneration, but it is unclear whether these regenerated IESCs are dependent on niche cells (Metcalfe et al, 2014). By exposing mice to three cycles of DSS, we created a chronic colitis in mice, in which chronic inflammation led to a large amount of regenerated colonic crypts (Wirtz et al, 2017). Interestingly, pYSTAT5 was found to be highly activated in those nascent colonic crypts, suggesting the protective role of pYSTAT5 in colonic IESC regeneration (Fig S1C). We next ask whether pYSTAT5 protects IESC regeneration by enhancing intestinal Paneth cell or colonic niche differentiation.

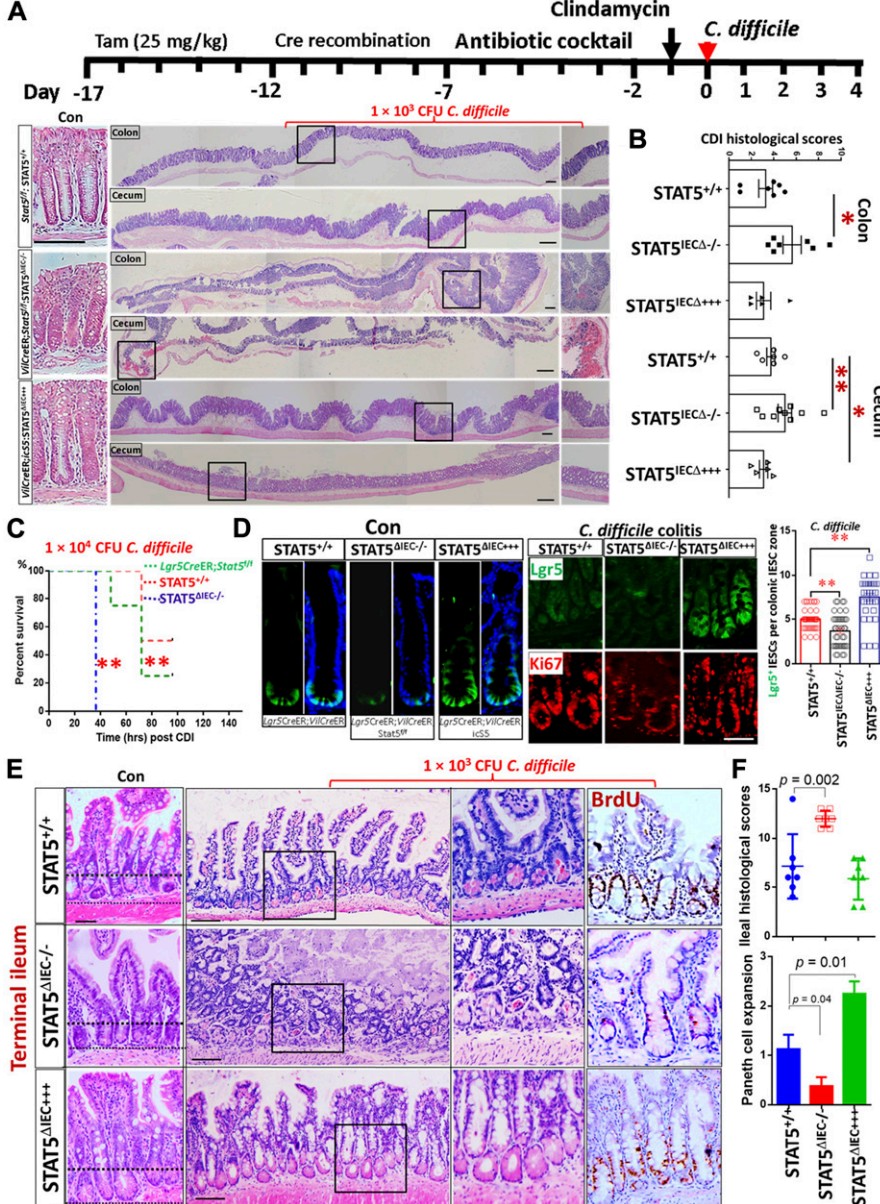

**Figure 1. Lack of pYSTAT5 predisposes mice to *C. difficile* ileocolitis.**
**(A, B)** STAT5$^{+/+}$, STAT5$^{\Delta IEC-/-}$, and STAT5$^{\Delta IEC+++}$ mice were inoculated with 1 × 10$^3$ CFU *C. difficile* for 4 d. Histopathology of colonic and cecal inflammation was scored. Results are expressed as mean ± SEM, n ≥ 5 mice per group, **$P < 0.01$ versus STAT5$^{+/+}$, *$P < 0.05$ versus STAT5$^{+/+}$. **(C)** Mice were inoculated with *C. difficile* at 1 × 10$^4$ CFU per mouse. Inducible depletion of STAT5 in IECs or IESCs significantly reduced survival following *C. difficile* infection. Survival was analyzed with Kaplan–Meier estimates, n = 7 mice per group, **$P < 0.01$ versus STAT5$^{+/+}$ mice. **(D)** *Lgr5Cre*ER;*VilCre*ER;icS5 mice were treated with *C. difficile*. Lgr5$^+$ IESCs were counted in 200 crypts in colonic mucosa, n ≥ 3 mice per group. Results are expressed as mean ± SEM, *$P < 0.05$ versus *Lgr5Cre*ER mice. Representative images of Lgr5$^+$ IESCs in control (Con) and *C. difficile* colitis are shown. **(E, F)** The severity of ileitis was scored as neutrophil infiltration, submucosal edema, IEC necrosis, and Paneth cell or goblet cell depletion. Paneth cell depletion or expansion was semi-quantitated in *C. difficile*-infected STAT5$^{+/+}$, STAT5$^{\Delta IEC-/-}$, and STAT5$^{\Delta IEC+++}$ mice. Histological scores show that STAT5$^{\Delta IEC-/-}$ mice display worse ileal inflammation than STAT5$^{+/+}$ mice, while STAT5$^{\Delta IEC+++}$ mice exhibit IEC protection and more regenerated BrdU$^+$ IECs. Results are expressed as mean ± SEM, *$P < 0.05$ versus STAT5$^{\Delta IEC-/-}$ mice, n ≥ 5 mice per group. All results are expressed as mean ± SEM, and *t* tests and ANOVA were used to compare the significance of a difference.

## Alteration of Paneth cell maturation and migration in IEC STAT5 mutant mice

Paneth cells produce the EGF, Wnt3, and Notch ligand Dll4 that are essential for Lgr5$^{hi}$ IESCs to establish a "3D" minigut, demonstrating that Paneth cells constitute the Lgr5$^{hi}$ IESC niche (Sato et al, 2011). Previously, we reported that IESCs require STAT5 signaling to regenerate damaged IECs (Gilbert et al, 2015). However, it is largely unknown how STAT5 controls IESC regeneration. Using inducible *VilCre*ER that can control levels of STAT5 expression in IECs in a Tam-inducible manner, we found that induced deletion of *Stat5* in IECs significantly reduced the numbers of Lysozyme (Lyso)$^+$ Paneth cells and the size of the niche (Figs 2A and S2A). However, Paneth cells are not completely disappearing, suggesting other

compensatory pathway activation or inhibition induced by STAT5 depletion (Heuberger et al, 2014). In contrast, STAT5 activation in IECs induced by 5 d of Tam treatment increased Lyso production (Fig 2B), numbers of Paneth cells (Fig 2A and C), and *Lyso*, *Muc2*, *Defensin1*, and *Defensin10* expression (Fig 2D). Interestingly, the activation of STAT5 led to a significant migration of Paneth cell from crypt bases towards villi, characterized by ectopic Lyso$^+$ cells above the crypt-villus junctions, more granules in the cytoplasm, and a larger niche volume than controls (Fig 2A and E). Of note, there are significantly increased numbers of Lyso$^+$AB$^+$ crypt IECs in the STAT5$^{\Delta IEC+++}$ compared with STAT5$^{+/+}$ mice but they are not on the villi (Fig S2B), suggesting that the activated STAT5 affect intermediate secretory progenitors towards crypt cell differentiation or dedifferentiation (Nusse et al, 2018; Schmitt et al, 2018). Perhaps

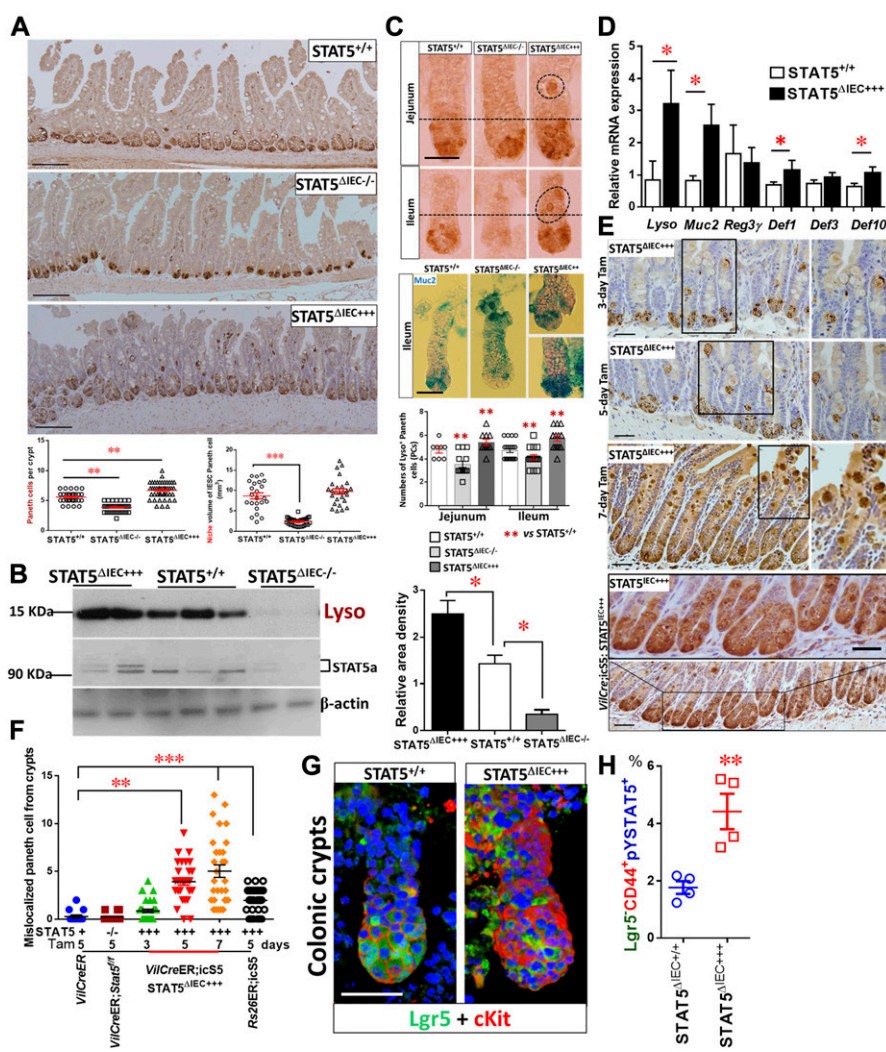

**Figure 2. STAT5 signaling regulates Paneth cells in a dose-dependent manner.**
*Stat5* or *icS5* floxed mice were crossed with *VilCre*ER. *VilCre*ER;*Stat5* or icS5 mice, and then treated with Tam for 3, 5, or 7 d followed by 5-d Cre recombination. **(A)** Paneth cells were determined with Lyso immunohistochemistry (IH) and IF staining. Average numbers of Lyso+ Paneth cells were counted in 200 crypts, and the size of Paneth niche was measured as volume with confocal microscopy (Fig S2A). **(B)** Intestinal crypts were isolated, total proteins were extracted, and immunoblotting was performed to determine Lyso and STAT5a protein expression. Densitometry was used to determine the expression of Lyso relative to β-actin. Results are expressed as mean ± SEM, n ≥ 5 mice per group. **(C)** Jejunal and ileal crypts were isolated and stained with Lyso IH and AB. Lyso+ crypt cells were counted in 200 isolated crypts. **(D)** Real-time PCR was performed to determine anti-microbial peptide expression in STAT5+/+ and STAT5ΔIEC+++ mice. **(E)** Ectopic Paneth cells were counted as the number of migrated Lyso+ IECs from crypt bases to villi in 3, 5, and 7 d Tam-treated *VilCre*ER or 5 d Tam-treated *Rs26Cre*ER or *VilCre*;icS5 mice. Scale = 200 μm. **(F)** Over 200 villi and crypts were counted. Results are expressed as mean ± SEM, n ≥ 5 mice per group. **(G)** Colonic crypts were double-stained with anti-Lgr5 (green) and anti-cKit (red), and 3D images were captured with confocal microscopy, n > 3 mice each group. **(H)** Colonic crypts were disassociated with TrypLE into IECs. Lgr5-GFP− and + IECs were then separated by gating with FACS. PE-Cy7-conjugated pYSTAT5 and APC-conjugated CD44 staining were used to quantitate pYSTAT5+Lgr5−CD44+ colonic crypt IECs, n = 4 mice per group, **$P < 0.01$ versus STAT5+/+. All results are expressed as mean ± SEM, and *t* tests and ANOVA were used to compare the significance of a difference.

more intriguing, the numbers of ectopic Paneth cells were correlated with the duration of Tam induction (Fig 2E and F), and persistent pYSTAT5 in IECs driven by *Vil*Cre (Fig S2C) led to broad Paneth cell expansion, metaplasia, crypt fission (Fig 2E), and the appearance of dysplastic crypts (Fig S2D). These data suggest that STAT5 is a key transcription factor that regulates Paneth cell differentiation.

Although Paneth cells are a defined niche of intestinal IESCs, they are not found in colon at homeostasis. There is a distinguished population of secretory IECs, they reside at the colonic crypt base to serve as colonic IESC niche, which can be labeled by cKit, Reg4, Muc2, and Alcian blue (AB), and this process requires Notch inhibition (Rothenberg et al, 2012; Sasaki et al, 2016). These secretory cells are currently referred to as colonic deep crypt secretory cells (DCSs) (Sasaki et al, 2016). We found that Ca-pYSTAT5 increased cKit+ crypt IECs and phospho-cKit (pcKit) in colonic crypt IECs (Figs 2G and S2E). Importantly, our FACS analysis showed that compared with controls, Ca-pYSTAT5 led to a population of colonic crypt cells: pYSTAT5+Lgr5−CD44+ cells (Fig 2H), which could have a direct role in protecting colonic IESCs from pro-inflammatory or *C. difficile* toxins.

Taken together, pYSTAT5 protects the host from *C. difficile* toxin injury, possibly by acting on intestinal Paneth cells and/or colonic DCS cells.

## Constitutive STAT5 signaling enhances lineage tracing at crypt bases

STAT5-dependent regeneration may be due to autonomous regulation of IESC progeny fates or secondary to other events, such as apoptosis or survival following injuries (Gilbert et al, 2015). To determine the primary effects of STAT5 on IESCs, we performed a lineage-tracing study. *Lgr5Cre*ER mice were first crossed with *Rosa26-stop-tdTomato* mice (RstdTomato) to generate a mouse line (hereafter called Lgr5-tdTomato). The Lgr5-tdTomato, a compound mutant model, permits tracing of Lgr5+ IESC progeny within intestines and robust labeling of daughter cells for visualization (Barker et al, 2007) (Fig 3A). Lgr5-tdTomato mice were then crossed with *icS5* floxed mice (thereafter called Lgr5-tdTomato;icS5). These compound mice allowed us to trace the effects of gain or loss of function of STAT5 in Lgr5 IESCs upon their

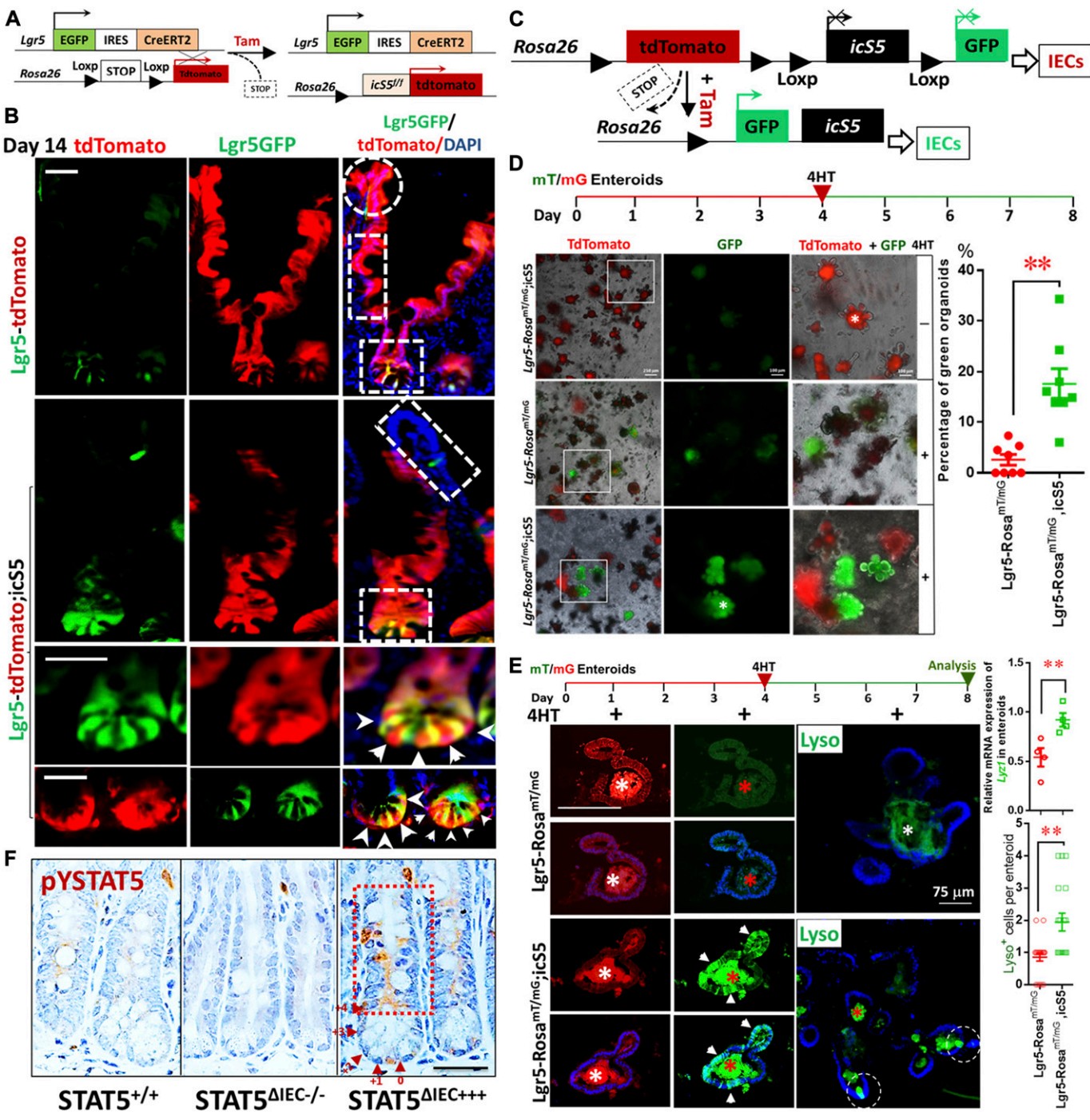

**Figure 3. pYSTAT5 enhances lineage tracing of crypt secretory cells.**
**(A)** Lgr5-tdTomato mice were generated for IEC lineage analysis with a single dose of Tam. **(B)** One dose of Tam (25 mg/kg) was given to Lgr5-RstdTomato;icS5 mice 14 d later. Intestinal lineage tracing shows that tdTomato-labeled Lgr5$^+$ progeny (red/green) are shown in the squares at crypt bases and Lgr5$^-$ and Paneth cells (red) are marked by arrowheads and counted (Fig S3C). tdTomato-labeled Lgr5$^-$(red) cells are shown in the rectangles on the villi, and the repopulated IECs are shown in the circle on the tip of the villus in the red ribbons. 200 crypts were counted, three mice per group. **(C)** Lgr5CreER mice were crossed with Rosa$^{mT/mG}$ mice to generate Lgr5-Rosa$^{mT/mG}$;icS5 mice. **(D)** Enteroids from Lgr5-Rosa$^{mT/mG}$;icS5 mice were differentiated, and then constitutive icS5 was induced with 4HT (200 nM) for 4 d. Enteroids prior to or after 4HT induction were imaged. The switch percentage from red to green enteroids was determined after 4HT induction in three experiments. **(E)** Enteroids prior to or after 4HT induction were fixed and sectioned. Images were collected. Arrowheads show green ribbon along villi. Lyso$^+$ Paneth cell-like cells were immunostained as the circles (green) and were quantitated. *Autofluorescence. Total RNA was extracted from the enteroids, and the expression of Lyz1 was quantitated by real-time PCR. Results are expressed as mean ± SEM, n = 4 or 5 wells of enteroids per group. **$P < 0.01$ versus Lgr5-Rosa$^{mT/mG}$. **(F)** Ileal sections were stained with pYSTAT5 IH (brown). n ≥ 5 mice per group, Scale = 200 μm. All results are expressed as mean ± SEM, and t tests and ANOVA were used to compare the significance of a difference.

progeny following a single dose of Tam induction (25 mg/kg, Figs 3A and S3A). After 1, 7, 15, 30, and 60 d following a single dose of Tam, lineage tracing demonstrated that the parallel red ribbons that contained all intestinal or colonic IEC types emanated from the crypt bottoms and ran up the side of adjacent villi in Lgr5-tdTomato mice (see squares at crypt and rectangles on villus, Figs 3B, S3B, and C). In contrast, the activated STAT5 in IESCs led to the red ribbons that remained at the intestinal and colonic crypt bases even after 15-d lineage tracing, showing significantly greater numbers of tdtomato-labelled Paneth cells and colonic Paneth-like cells at the crypt bases compared with controls (see squares at crypts, Figs 3B, S3B, and C); these stained IEC lineages climbed significantly slower along the intestinal villi compared to those in their littermate controls 15, 30 (data not shown), or 60 d later (see rectangles on villi, Figs 3B and S3C). Lgr5CreER mice were then crossed with Rosa26-stop-LacZ mice (LacZ) to generate a mouse line (thereafter called Lgr5-LacZ), which can be used to demonstrate the regulatory genes of IESC progeny fates (Fig S4A) (Barker & Clevers, 2010). Interestingly, following a single dose of Tam, Lgr5-LacZ;Stat5 mice displayed a reduced LacZ staining and decreased tracing presence at the intestinal or colonic bases (Fig S4B–D), indicating that STAT5 is an intrinsic signaling that is required for IESC self-renewal.

$Rosa^{mT/mG}$ mice present a conditional fluorescent reporter mouse model ($Rosa^{mT/mG}$), in which the fluorescent protein-encoding transgenes were rearranged following Tam-mediated Cre-recombinase and transgene expression of red-tdTomato fluorescent protein was converted to the expression of green-GFP (Fig 3C) (Muzumdar et al, 2007). By crossing $Rosa^{mT/mG}$ mice with Lgr5CreER-icS5 mice (hereafter called Lgr5-$Rosa^{mT/mG}$;icS5), we generated a line of mice with Tam-inducible constitutively active STAT5 (Fig 3C). Subsequently, we differentiated 14-d enteroids and maintained part of these in culture medium with 200 mM 4-hydroxytamoxifen (4HT) for 4 d (Gilbert et al, 2015). Concomitant with in vivo lineage tracing, our in vitro lineage study showed that after 4HT-induction, red-tdTomato fluorescence in Lgr5-$Rosa^{mT/mG}$;icS5 enteroid was largely absent, while green-GFP cells, a large portion of which were stained as Lyso⁺ Paneth cells or Sox9⁺ cells, were highly and rapidly enriched compared to 4HT-treated Lgr5-$Rosa^{mT/mG}$ controls. These enteroids also expressed the higher levels of Lyz1 or Sox9 than controls (Figs 3D, E, S4E, and F). Taken together, these data indicate that STAT5 signaling regulates IESC progeny differentiation and more specifically that pYSTAT5 in IESCs enhances IESC lineage tracing to increase crypt Paneth cell differentiation, while loss of Stat5 leads to reduced IESC self-renewal and stemness. Interestingly, our compound mouse models showed that constitutively activated STAT5 in IECs increased pYSTAT5 in the transit-amplifying (TA) IECs and crypt base IECs (see rectangle and arrowheads, Fig 3F), some of which were co-localized with nucleic Ki67 in TA IECs (see circles in TA zone, Fig S4G). This observation suggests that these pYSTAT5⁺Lgr5lowKi67⁺ IECs in the TA zone may be a progenitor population determined by cytokines, hormones, or growth factors that steer pYSTAT5 signaling from Lgr5 IESCs (Basak et al, 2014). Thus, we next determined the functions of these pYSTAT5⁺Lgr5lowKi67⁺ IECs and the fate of their progeny as induced by pYSTAT5.

## pYSTAT5 restricts Lgr5lowKi67⁺ IESCs to give rise to pYSTAT5⁺Lgr5⁻CD24⁺ Paneth cells

Clevers' group reported that the Lgr5lowKi67⁺ IESC population above Paneth cells expresses high levels of Stat5a gene in association with significantly increased gene signatures of secretory lineages after induced activation (Basak et al, 2014). These secretory cells induced from Lgr5lowKi67⁺ IESCs reside at the crypt base to serve as an IESC niche, which can be labeled by Lyso, c-Kit, CD24, CD44, and AB staining (Rothenberg et al, 2012; Sasaki et al, 2016). These reports suggest that Stat5 may be required for Lgr5lowKi67⁺ IESCs to differentiate into Paneth cells at crypt bases, providing the most critical regional immunity against microbial invasion and inflammatory insults (Mowat & Agace, 2014). After 5-d Tam induction, we found that activated STAT5 significantly increased pYSTAT5⁺Lgr5⁻CD24⁺ intestinal crypt IECs detected in gate P1 by FACS tracing experiments (Figs 4A and S5A). On the contrary, the depletion of Stat5 led to a pronounced reduction of Lgr5⁻CD24⁺ IECs detected in gate P2 (Fig 4A). These results are consistent with our above observations which showed that activated STAT5 increased Paneth cells at the crypt bases (Fig 2A). Intriguingly, using the dissociated IECs from directly isolated crypts, the FACS analysis revealed that the activated STAT5 increased Lgr5hi:CD24⁺ cell doublets (Figs 4B and S5B). This result is consistent with our confocal microscopy results, showing the increased numbers of Lgr5-GFP and CD24 co-immunostaining doublet in STAT5ΔIEC⁺⁺⁺ mice (see circles, Fig 4C). Thus, STAT5 activation increases IESC niche cells.

Single-sorted Lgr5hi crypt base columnar cells (CBCs) rarely generate miniguts, whereas CBC:Paneth cell doublets robustly grow into a minigut (Sato et al, 2011). Thus, we grew enteroids containing Lgr5hi CBCs and niche cells from Lgr5CreER;VilCreER,icS5 mice and performed an injury study with irradiation to investigate the effects of STAT5 activation upon the regenerative capacity of Lgr5-GFP⁺ enteroids (Fig 4D). The activated STAT5 increased Lgr5⁺ enteroid regeneration as characterized by significantly more remaining Lgr5 GFP⁺ buds in the enteroids compared with irradiation-treated controls (Lgr5CreER;VilCreER;icS5 versus Lgr5CreER;VilCreER mice, Figs 4D and S5C), higher proliferation of Lgr5hiKi67⁺pYSTAT5⁺ IESCs, and non-significant change of Lgr5lowKi67⁺pYSTAT5⁺ IESCs under basal conditions (Fig 4E and F). Interestingly, irradiation reduced Lgr5lowKi67⁺pYSTAT5⁺ IESCs while increased Lgr5hiKi67⁺pYSTAT5⁺ IESCs in the control enteroids compared with basal conditions. In contrast, the activated STAT5 induced more Lgr5lowKi67⁺pYSTAT5⁺ IESC regeneration after irradiation compared to irradiation-treated control enteroids (Fig 4F). Consistently, the activated STAT5 increased the resistance of colonoids to the radiation-induced injury, as shown by more Lgr5-GFP buds in STAT5-activated colonoids compared with the relatively less buds in controls 6 d post-radiation (Fig 4G). Stem cell factor (SCF), a cKit ligand, is known to act as an agonist of JAK2-STAT5 signaling (Nakai et al, 2008). SCF is highly expressed in the subepithelial stromal cells during gut injury (Schmitt et al, 2018). These secreted SCF could be the source to induce pYSTAT5 in cKit⁺ DCSs. Consistently, we found that SCF significantly promotes colonoid proliferation at 20 ng/ml (Fig S5D and E). Therefore, Ca-pYSTAT5 stimulates colonoid proliferation possibly by increasing SCF-activated cKit colonic IECs. Together, these data suggest that the STAT5 activation protects IESC

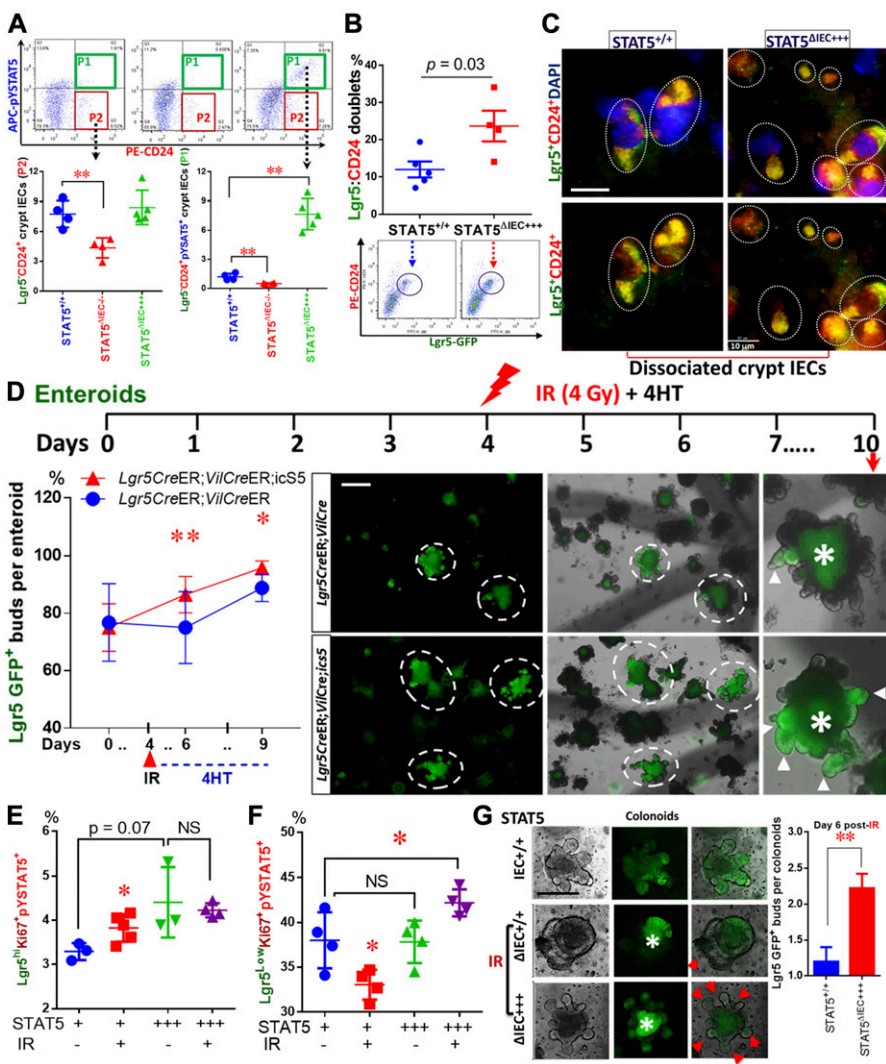

Figure 4. pYSTAT5 restricts Lgr5$^{low}$Ki67$^+$ IESCs to give rise to pYSTAT5$^+$Lgr5$^-$CD24$^+$ Paneth cells.
Intestinal crypts were extracted from *Lgr5Cre*ER; *VilCre*ER;Stat5 or icS5 and dissociated with TrypLE into IECs. Lgr5$^-$, low and high IECs were then separated by gating with FACS (Fig S5A). **(A)** APC-conjugated pYSTAT5 and PE-conjugated CD24 staining were used to quantitate pYSTAT5$^+$Lgr5$^-$CD24$^+$ cells (P1) or Lgr5$^-$CD24$^+$ Paneth cells (P2). **(B)** Lgr5$^{hi}$:CD24$^+$ doublets in the dissociated IECs were determined by FACS. n = 4 or 5 mice per group, **$P$ < 0.01 versus STAT5$^{+/+}$. **(C)** The dissociated IECs were immunostained with CD24, and co-localization of Lgr5 and CD24 was determined with a confocal microscope. Lgr5:CD24 doublets are shown as circles. **(D)** Lgr5-GFP crypts from *Lgr5Cre*ER or *Lgr5Cre*ER; *VilCre*ER;icS5 mice were employed for IESC culture for 4 d, and pYSTAT5 was inducibly activated by 4HT (200 nM) after one dose of γ-irradiation (IR). The numbers of Lgr5-GFP buds were counted in each well, and budding curve was created 6 d after 4HT treatment. **$P$ < 0.01 and *$P$ < 0.05 versus controls without 4HT induction. Representative images of Lgr5GFP$^+$ buds (arrows) in the enteroids (circles) are shown, *Autofluorescence. **(E, F)** Lgr5-GFP enteroids were dissociated into IECs. Lgr5$^{hi}$Ki67$^+$pYSTAT5$^+$ (E) and Lgr5$^{low}$Ki67$^+$pYSTAT5$^+$ (F) were determined. *$P$ < 0.05 versus enteroids from *Lgr5Cre*ER; *VilCre*ER, #$P$ < 0.05 versus enteroids from IR-treated *Lgr5Cre*ER; *VilCre*ER, n = 4–5 mice per group. **(G)** Colonic crypts were isolated and differentiated into colonoids. These colonoids were induced by 4HT for 4 d and then irradiated at 4 Gy for 10 min. Lgr5 (green) buds are shown; arrowheads indicate crypt budding. GFP crypt buds were counted per colonoids (n ≥ 20) from each of six wells from three independent experiments. All results are expressed as mean ± SEM, and *t* tests and ANOVA were used to compare the significance of a difference.

regeneration, possibly by increasing Lgr5$^{low}$Ki67$^+$IESC activation or differentiation towards niche cells.

## pYSTAT5 protects Lgr5$^{hi}$Ki67$^+$ IESCs from inflammatory cytokine or toxin-induced IEC injuries by enhancing niche function

In vivo studies showed that inflammatory cytokines and toxins may induce IESC apoptosis or cause Paneth cell degeneration (Farin et al, 2014; Nakanishi et al, 2016); however, whether inflammatory bacterial toxins directly injure IESCs remains unclear. Constitutive pYSTAT5 increased IESC regeneration (Gilbert et al, 2015); thus, we treated enteroids from *Rs26*tdTomato;icS5 with TNFα (100 ng/ml), toxin A (TcdA, 0.5 μg/ml), or different doses of toxin B (TcdB; 12, 24, or 36 ng/ml). Based on bud numbers (arrows), we categorized enteroid morphology as 1 bud (Org$^1$), 2 buds (Org$^2$), greater than 3 buds (Org$^{3+}$) and spheres (circles) (Figs 5A and S6A). We found that TNFα stimulation increased sphere numbers and size, whereas it reduced budding numbers and enteroid survival (Figs 5A, S6A, and B). For example, TNFα diminished enteroids with over 3 buds (Org$^{3+}$) (Fig 5A). In contrast, the activation of STAT5 decreased

sphere numbers and increased buds in the TNFα-treated enteroids (Fig 5A), suggesting that activated STAT5 may counteract TNFα-induced inhibition of IESC niche regeneration. We next exposed enteroids to different doses of TcdB. The budding curves were created to represent the average buds per enteroid at different time points (n ≥ 20 enteroids per well, six wells per mouse, and three mice per group). ANOVA was used to compare the difference between groups. We found that a high dose of TcdB (36 ng/ml) led to a significant reduction in crypt budding in the Lgr5-GFP enteroids (Fig 5B-I, **$P$ < 0.01 versus control). In contrast, inducible activation of icS5 partially restored enteroid proliferation and viability from TcdB cytotoxicity, which were determined by the number of Lgr5$^+$ buds per enteroids and Methylene Blue staining (Fig 5B-II and C, *$P$ < 0.05 versus Con + 30 ng/ml TcdB). These data suggest that the forced pYSTAT5 protect enteroids from TcdB-induced inhibition of IESC regeneration or niche activity.

Furthermore, using FACS, we found that TcdB not TcdA treatment led to increased Lgr5$^{hi}$Ki67$^+$ apoptosis (Fig 5D) and reduced Lgr5$^{hi}$Ki67$^+$ proliferation (Fig 5E). In contrast, the activation of STAT5 reduced Lgr5$^{hi}$ or Lgr5$^{low}$BrdU$^+$ IESC apoptosis induced by TcdB (Fig

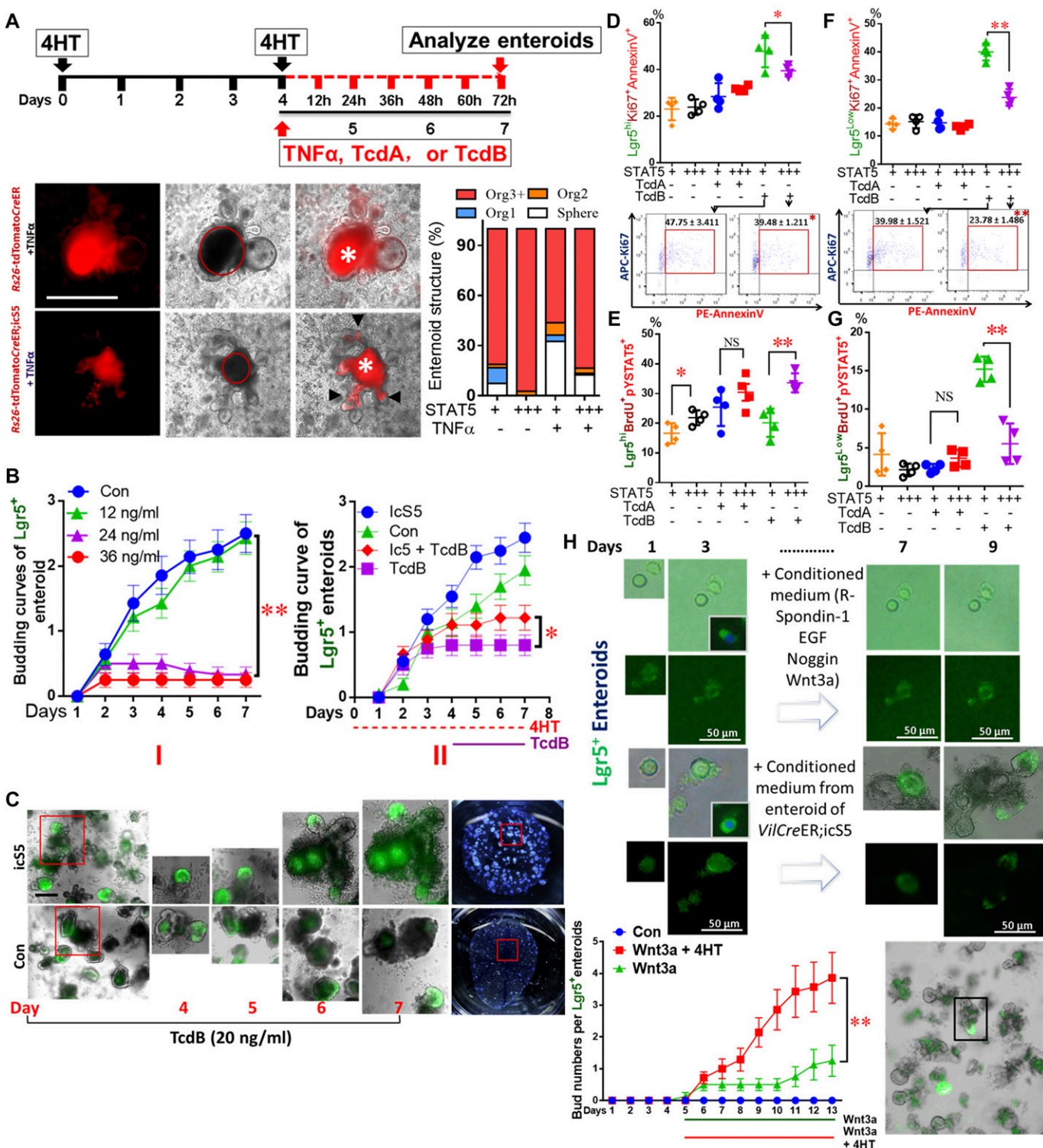

**Figure 5. pYSTAT5 protects Lgr5hiKi67+ IESCs from inflammatory cytokine- or toxin-induced IEC injuries by enhancing niche function.**
**(A)** Enteroids were cultured. pYSTAT5 was induced with 4HT for 4 d, then exposed to TNFα, TcdA, or TcdB for 3 d. Enteroids were cultured from intact intestinal crypts dissociated from *Rs26-tdTomatoCre*ER and *Rs26-tdTomatoCre*ER;icS5 mice. Based on the number of buds, enteroid morphology was categorized as follows: 1 bud (Org[1]), 2 buds (Org[2]), greater than 3 buds (Org3+) and no buds (Sphere). Results were expressed as mean ± SEM, n = 4 or 5 mice per group. *Autofluorescence. **(B)** Enteroids from *Lgr5Cre*ER;*Vilcre*ER and *Lgr5Cre*ER;*Vilcre*ER;icS5 were cultured, then were treated with TcdB (I). After icS5 was induced on day 4, enteroids were treated with 30 ng/ml TcdB for 3 d (II). **(C)** Representative images of Lgr5+ enteroids from days 4 to 7 with or without TcdB or icS5 induction are shown. These treated enteroids were then dissociated for FACS. **(D, E)** APC-conjugated AnnexinV and PE-conjugated pYSTAT5 were used to determine the effects of pYSTAT5 upon Lgr5hiKi67+ proliferation or Lgr5lowKi67+ survival. **(F, G)** APC-conjugated BrdU and Percep-conjugated pYSTAT5 were used to determine Lgr5hiBrdU+pYSTAT5+ or Lgr5lowBrdU+pYSTAT5+ IESCs, which were

5D and F) and stimulated Lgr5$^{hi}$BrdU$^+$ IESC proliferation (Fig 5E) and repressed Lgr5$^{low}$BrdU$^+$IESC proliferation (Fig 5G). Taken together, these data suggest that pYSTAT5 prevents IESCs from TcdB inhibition by protecting Lgr5$^{hi}$BrdU$^+$ IESCs or enhancing switching from Lgr5$^{hi}$BrdU$^+$ →to Lgr5$^{low}$BrdU$^+$ IESCs for niche regeneration.

Finally, utilizing the conditioned feeding medium made from WT or *VilCre*ER;icS5 ileal enteroids, we transferred the conditioned feeding medium into sorted Lgr5$^{hi}$ IESCs (Fig S6C). We performed a "medium replacement" experiment to study whether the factors secreted by icS5-activated enteroids can facilitate Lgr5$^{hi}$ IESCs to differentiate as enteroids (Fig 5H). Incubation with medium from icS5 ileal enteroids (Wnt3a + 4HT) significantly increased Lgr5$^{hi}$ IESC proliferation compared with conditioned medium made from WT enteroids (*P* < 0.01 versus Wnt3a group), indicative of a role of pYSTAT5 or pYSTAT5-derived factors secreted by icS5 enteroids on the niche. In contrast, by counting the percentage of the final survival enteroids versus initial grown enteroids in the presence or absence of TNF-*α* (Schewe et al, 2016), our results showed that the reduced STAT5 by partially depleting *Stat5* significantly impaired the regenerative capacity of enteroids, displaying a compromised organoid formation manifested by the diminished bud numbers, enlarged spheroid size, and reduced survival of organoids, the phenotypes of which are significantly exaggerated by the presence of TNF-*α* (Fig S6A and B). Taken together, our data indicated that pYSTAT5 may restrict Lgr5$^{low}$Ki67$^+$ IESCs to give rise to pYSTAT5$^+$ Lgr5$^-$CD24$^+$ Paneth cells, which protect Lgr5$^{hi}$ IESCs from inflammatory cytokines or enteric toxin–induced IESC injuries, whereas reduced pYSTAT5 impairs the regenerative capacity of the enteroids in homeostasis and inflammation conditions.

## Constitutive pYSTAT5 enhances heterogeneous monolayer integrity and increases Lysozyme production in human intestinal organoids (HIOs)

We generated iPSCs stably expressing wild-type (WT-GFP), inducible active forms of STAT5A (STAT5a-ER) and inducible constitutively active STAT5 (icS5-ER) (Gilbert et al, 2015). In these iPSCs, pYSTAT5 can be inducibly or persistently activated in a dose-dependent manner (Fig S7A). Transduced iPSCs are able to be differentiated into HIOs (Spence et al, 2011). These HIOs were cultured under conditions that promote intestinal growth, morphogenesis, and cyto-differentiation into functional intestinal cell types including enterocytes, goblet cells, Paneth cells, and enteroendocrine cells (Fig 6A) as previously reported (Spence et al, 2011). iPSC-derived HIOs contain undifferentiated progenitors or fetal-like stem cells that could maintain HIO regeneration during injury (Spence et al, 2011; Watson et al, 2014). Therefore, HIOs represent a novel human primary cell model system that can be used for studying IESC regeneration-dependent IEC repair as summarized by the Wells group (Sinagoga & Wells, 2015). Our data showed that 2-mo HIOs contained heterogeneous monolayers with fully differentiated IECs

(Paneth, goblet, enterocyte, and enteroendocrine cells) with intercellular junctional barriers (Figs 6A and S7C) (Spence et al, 2011; Watson et al, 2014). We next differentiated lentivirus-transduced iPSCs into HIOs after 2 mo in culture and assessed HIO lineage integrity by measuring the efflux of microinjected fluorescein isothiocyanate-dextran 4 (FD4) and transepithelial electrical resistance (TEER) (Figs 6B and S7B). We found that TcdB impaired HIO lineage integrity during early time treatment (within 6-h of treatment) but not TcdA, while Cytomix (IFNγ and TNFα) led to disruption of HIO cell integrity and leaky junctions after 24 h of treatment. In contrast, persistent pYSTAT5 induced by icS5-ER, but not transient pYSTAT5 induced by STAT5a-ER, restored HIO lineage integrity following Cytomix treatment (Fig 6B). This persistent pYSTAT5 also resulted in an up-regulated *β*-catenin expression, increased LGR5$^+$ IECs, expression of inter-cellular junctional proteins (E-cadherin, occludin, ZO-1, and ZO-2), and Lysozyme expression and inhibited quiescent IESC markers Bmi1 and Dclk1 (Figs 6C and S7C). These data suggest that icS5-induced persistent pYSTAT5 expression repairs leaky HIOs by enhancing HIO de novo differentiation to promote heterogeneous monolayer maturation.

## Constitutive pYSTAT5 induces de novo niche differentiation to protect IESC regeneration

Transplantation of HIOs into the kidney capsules of immunodeficient NOD-Scid IL2Rynull (NSG) mice has previously been used to evaluate functional engraftment of tissues, IESC-regenerative repair, and IEC lineage differentiation (Watson et al, 2014). We implanted HIOs that were differentiated for 2 mo into the kidney capsule (Fig 6D) and allowed them to develop into intestinal tissue according to the published protocols (Spence et al, 2011; Watson et al, 2014) (Fig 6E). We found that inducible activation of icS5 significantly increased Ascl2$^+$ and Lgr5$^+$ CBC proliferation (Fig 6E), suggesting a direct effect of persistent pYSTAT5 signaling on increasing numbers of Lgr5 IESCs. Using a knock-in and Tam-inducible Cre recombinase to facilitate tracing of the lineage of a single human LGR5 stem cell during HIO development (Fig 6F) we found that the activation of icS5 yielding a persistent pYSTAT5 in transplanted HIOs significantly enhanced LGR5 GFP lineage tracing (Fig 6G), and Lyso$^+$ Paneth and Muc2$^+$ goblet cells at crypt bases in the transplanted HIO-derived intestinal tissues (Fig 6H). After exposing the transplanted mice to 5 d of Tam following 5 d of Cre recombination, we then exposed the transplanted NSG mice to one dose of 12 Gy irradiation and tested the function of engrafted IESC activation for regenerative repair 3 d post-irradiation. Persistent pYSTAT5 resulted in well-preserved regenerated crypts compared with transient pYSTAT5, which exhibited few nascent crypts 3 d post-irradiation (Fig 6I). Taken together, our data strongly suggest that the activation of icS5 increases IESC regenerative capacity, possibly through de novo enhancement of Paneth cell niche formation to maintain IESC regeneration.

---

used to quantitate Lgr5$^{hi}$ or Lgr5$^{low}$ IESC proliferation, n = 4–5 mice per group. **(H)** Lgr5$^{hi}$ IESCs were sorted from *Lgr5CreER* mice or *Lgr5Cre*ER;icS5 mice and enteroids were cultured. Conditioned medium was generated from enteroids from *VilCre*ER;icS5 mice. Lgr5$^{hi}$ IESCs were sorted and grown into enteroids to compare the effects of the conditioned medium versus regular medium on Lgr5$^{hi}$ single cell growth and enteroid differentiation. All results are expressed as mean ± SEM, and *t* tests and ANOVA were used to compare the significance of a difference.

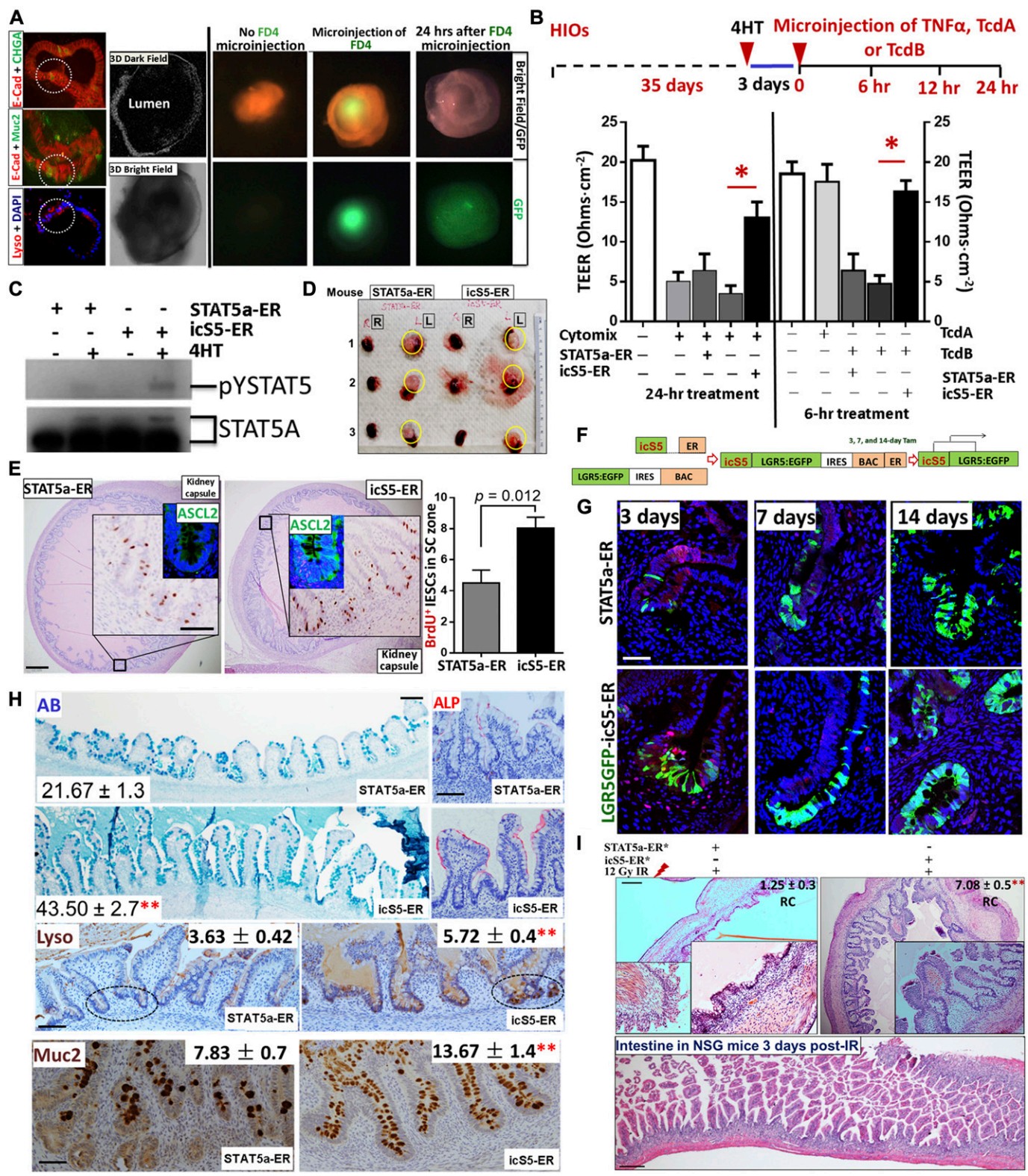

**Figure 6. Constitutive pYSTAT5 induces de novo niche differentiation and protects IESC regeneration.**
iPSCs or LGR5:eGFP BAC reporter iPSCs were transduced with a lentiviral GFP construct for STAT5a-ER and icS5-ER mutant. HIOs were in vitro matured for 35 d. **(A)** Transduced HIOs were imaged under 3D in vitro culture systems with dark-field and bright-field 3D confocal deconvolution microscopy. IEC types in HIOs were determined by double-IF, E-Cadherin (E-Cad) + Chromogranin A (CHGA) or E-Cad + Muc2, and Lyso IF staining. **(B)** Transduced HIOs were then microinjected 250 ng FD 4 (FD4) with 0.25 ng TNFα (100 ng/ml) or 25 ng TcdA or 50 ng TcdB after 3-d 4HT induction. HIO integrity was determined as the TEER after 24-h TNFα or 6-h TcdA or

## STAT5 activation regulates Wnt-β-Catenin-Sox9 signaling axis to sustain niche cell differentiation in a stepwise manner

To determine the potential molecular mechanisms by which STAT5 signaling controls niche cell homeostasis, we isolated total RNA from intact intestinal crypts of STAT5$^{+/+}$, STAT5$^{ΔIEC-/-}$, and STAT5$^{ΔIEC+++}$ mice and performed RNA sequencing (RNA-seq). The sample correlation matrix and hierarchical clustering based on the correlation showed that the genes between STAT5$^{+/+}$, STAT5$^{ΔIEC-/-}$, and STAT5$^{ΔIEC+++}$ mice were clustered separately. Furthermore, the altered gene signatures of intestinal crypt IECs were correlated with *Stat5* depletion or pYSTAT5 activation ($P < 0.05$ and fold change >2 in at least one pairwise comparison of 691 genes, Fig 7A). Scatter plots of the log2 (fragments per kilobase of exon model per million reads mapped (FPKM)) counts of genes with a cutoff of 1.0 for the absolute value of log2 (ratio) of expression levels showed greater than onefold significant changes in differential gene expression between STAT5$^{ΔIEC-/-}$ and STAT5$^{+/+}$ mice and between STAT5$^{ΔIEC+++}$ and STAT5$^{+/+}$ mice (Fig 7B). Venn diagrams showed that shared and unique genes were differentially expressed in STAT5-deficient or pYSTAT5-activated crypt IECs. Of these genes, 184 were up-regulated genes in STAT5$^{ΔIEC+++}$ mice compared with STAT5$^{+/+}$ mice, and 52 were down-regulated genes in STAT5$^{ΔIEC-/-}$ mice compared with STAT5$^{+/+}$ mice. Furthermore, we observed 54 down-regulated and 85 up-regulated genes specific to STAT5$^{ΔIEC-/-}$ mice, and 122 up-regulated and 126 down-regulated genes specific to STAT5$^{ΔIEC+++}$ mice (Fig S8A).

Gene ontology enrichment analysis of these identified genes indicated that pYSTAT5 activation in IECs up-regulated intestinal innate immunity, cytokine production, IFNγ, and autophagy, and down-regulated chromatin silencing, susceptibility to infection, and DNA damage–induced senescence (Fig S8B). These findings indicate that pYSTAT5 activation enhanced intestinal secretion and innate immune reaction to acute infection. In contrast, the depletion of *Stat5* exaggerated immune response, inflammation, and wounding responses and decreased Th2-associated cytokines IL-10, IL-7, and thyroid peroxidase and IL-3 anti-inflammatory JAK-STAT signaling (Fig S8C), suggesting the suppression of Th2 immune induction or crypt cell activation. These identified differentially expressed genes were then subjected to Ingenuity Pathway Analysis. Inflammatory pathways such as TNFα, IFNγ, IL-4, IL-13, and IL-10 or IL-22 inflammatory or anti-inflammatory pathways were enriched in STAT5$^{ΔIEC-/-}$ and STAT5$^{ΔIEC+++}$ compared with STAT5$^{+/+}$ mice (Fig S8D). These data suggest that STAT5 proteins in crypt IECs are essential for intestinal immune and inflammatory responses. Furthermore, these data suggest that STAT5 signaling regulates intestinal inflammation responses to shape innate immune development.

Next, we studied the genes and pathways downstream of STAT5 signaling to elucidate the molecular mechanisms of STAT5 restriction of crypt Paneth cell fates from progenitors. Constitutive Wnt signaling allocates Paneth cells along the crypt base to the villus (van Es et al, 2005). Intriguingly, toxin B was found to compete with Wnt signaling to bind Frizzled (FZD) family of receptors to mediate colonic epithelial injury (Chen et al, 2018). These results indicated that *C. difficile* toxins may cause intestinal injury by impairing Wnt signaling-dependent Paneth cells. Using hierarchical clustering of averaged normalized expression, we found that inducible activation of intestinal pYSTAT5 significantly regulated the key Wnt signaling pathway genes. These genes, Fzd5-Ctnnb1-Tcf4 (Tcf7L2)-Sox9, Gfi1, and EphB2/3, were differentially regulated in STAT5$^{ΔIEC-/-}$ or STAT5$^{ΔIEC+++}$ mice compared with STAT5$^{+/+}$ mice (Fig S8E). Interestingly, other key genes that regulate Paneth cell functions were also significantly altered, such as anti-microbial *Defa5*, metabolic and inflammatory *Lcn2* (Vijay-Kumar et al, 2010), autophagy-related *Xbp1* (Kaser et al, 2008), and ER protein *Agr2* (Zhao et al, 2010) that regulate Paneth cell functions were also significantly altered ($P < 0.001$, STAT5$^{ΔIEC+++}$ and STAT5$^{ΔIEC-/-}$ versus STAT5$^{+/+}$ mice, Fig S8E). Consistently, we found that pYSTAT5 in STAT5$^{ΔIEC+++}$ mice markedly increased nucleic β-catenin and Sox9 in IECs at the crypt bases compared to controls (Fig 7C and D). We then measured the induction of pYSTAT5 levels in icS5-transduced HIOs with different doses of 4HT. Our results showed that pYSTAT5 induction in HIOs appeared 4HT dose-dependent from 10 to 200 nM and enhanced β-catenin expression (Fig 7E). A stepwise β-catenin-Tcf4-Sox9 pathway is the key pathway for determining Paneth cell commitment in the crypt progenitor fate towards niche differentiation (Mori-Akiyama et al, 2007). Accordingly, our data suggest that pYSTAT5 activation is essential for the differentiation of intestinal niche cells, likely Paneth cells. STAT5 may be a critical lineage-specific transcription factor that regulates intestinal niche differentiation and regeneration as well as anti-inflammatory or infection functions.

## Discussion

*C. difficile* infection causes diarrhea and pseudomembranous colitis, which are strongly associated with increased inflammation severity in IBD (Berg et al, 2013). While it remains unknown whether *C. difficile* infection has a causal link with IBD (Monaghan et al, 2015), a recent multi-center study defined a baseline prevalence of 7.5% for *C. difficile* infection amongst pediatric IBD patients compared with only 0.8% for controls with celiac disease (Martinelli et al, 2014).

---

TcdB treatment. Results are expressed as mean ± SEM. *$P < 0.05$ versus control HIOs. n ≥ 4 HIOs per group. **(C)** NEs were extracted from single HIOs, and pYSTAT5 and STAT5 were measured using immunoblotting. n ≥ 4 HIOs per group. **(D)** Transduced HIOs were matured and then transplanted beneath the kidney capsule of NSG mice. Tam induction was performed for 5 d 1-mo post-engraftment. **(E)** Proliferation of transplanted IESCs was determined with anti-Ascl2 and BrdU IH in the presence and absence of icS5 activation, n = 5 per group. **(F, G)** LGR5:eGFP BAC reporter iPSCs were transduced with lenti-viral STAT5a-ER or icS5-ER. 3, 7, or 14 d after one dose of Tam, transplanted mice were euthanized. Transplanted HIOs were stained with GFP IF, and IESCs were labeled with LGR5-GFP and EdU. icS5 activation increased LGR5$^+$ IESCs and enhanced LGR5 lineage tracing compared with that in STAT5a-ER. n ≥ 3 mice per group. **(H)** Transplanted HIOs were stained with AB, Alkaline phosphatase (ALP), Lyso, and Muc2 IH. AB$^+$, Lyso$^+$, or Muc2$^+$ crypt cells were counted as average numbers per crypt. 100 well-orientated crypts were chosen from five mice per group, **$P < 0.01$ versus STAT5a-ER. **(I)** Some of the transplanted mice were subjected to 12-Gy irradiation. Regenerated crypts (RC) were counted in the transplanted HIOs. Inserts are the images at a higher-magnification. Intestines from the irradiated mice show no RC. **$P < 0.01$ versus STAT5a-ER, n ≥ 5 mice per group, scale = 200 μm. All results are expressed as mean ± SEM, and t tests and ANOVA were used to compare the significance of a difference.

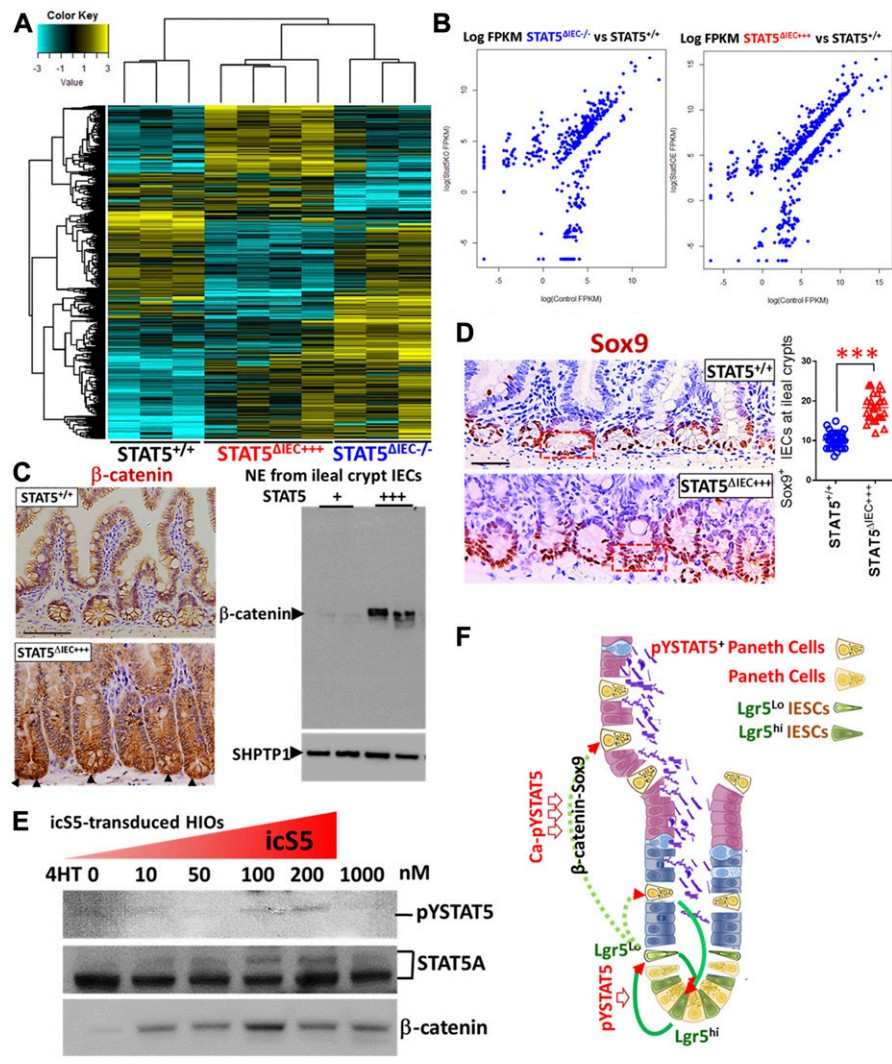

**Figure 7. STAT5 activation enhances IESC niche differentiation by regulating the IESC transcriptome.**
**(A)** Hierarchical clustering of 691 genes with ANOVA $P <$ 0.05 and fold change > 2 in at least one pairwise comparison between STAT5$^{+/+}$, STAT5$^{\Delta IEC-/-}$, and STAT5$^{\Delta IEC+++}$ mice, using Pearson's centered distance metric and average linkage rule. **(B)** Scatterplots of log (FPKM) of differentially regulated genes in STAT5$^{\Delta IEC-/-}$ and STAT5$^{\Delta IEC+++}$ mice when compared to STAT5$^{+/++}$ mice. **(C)** $\beta$-catenin was determined by IH and immunoblotting with NE from intestinal crypts. Results are expressed as mean ± SEM, n ≥ 3 mice per groups, and $t$ tests were used to compare the significant difference. **(D)** Sox9 was immunostained and quantitated in the 200 crypts. Results are expressed as mean ± SEM, n ≥ 3 mice per groups, and $t$ tests were used to compare the significant difference. **(E)** icS5-ER-transduced HIOs were in vitro matured for 35 d. icS5 activation in HIOs was induced with different doses of 4HT (0, 10, 50, 100, or 200 nM or 1 mM) for 72 h. pYSTAT5, STAT5A, and $\beta$-catenin were determined by immunoblotting. This experiment was repeated three times. **(F)** Hypothesis model: pYSTAT5 amplifies Lgr5$^{hi}$ →to Lgr5$^{Low}$Ki67$^{+}$IESCs; Ca-pYSTAT5 promotes Lgr5$^{low}$Ki67$^{+}$ IESCs to give rise to a sublineage of crypt cells, intestinal pYSTAT5$^{+}$Lgr5$^{-}$CD24$^{+}$ Lyso$^{+}$ Paneth cells by activating Wnt/$\beta$-catenin/Sox9 pathway.

Furthermore, patients with *C. difficile* infection were more likely to subsequently require escalation of medical therapy and hospitalization during follow-up. Meta-analysis showed that *C. difficile* infection was a significant risk factor for colectomy among patients with IBD, indicating that *C. difficile* infection may greatly reduce mucosal healing in IBD (Chen et al, 2017). Neither antibiotics, proton pump inhibitors, hospitalization, nor IBD therapies were associated with *C. difficile* infection (Martinelli et al, 2014). Interestingly, *C. difficile* can maintain colonization in the intestines, without inducing enteritis or colitis, suggesting an essential role of innate host defenses in the protection from *C. difficile* infection to colitis (Monaghan et al, 2015). Importantly, Paneth cells maintain host–microbiota homeostasis (Zhang et al, 2015a), and as such, genetic defects in Paneth cells induce intestinal inflammation and increase susceptibility to both IBD (Wehkamp et al, 2007) and *C. difficile* colitis (Giesemann et al, 2008). Together, these reports suggest a crucial role for host factors such as genetic variants, IESC regeneration, or Paneth cell specialization in the *C. difficile*-associated comorbid conditions, such as IBD. Therefore, targeting endogenous IEC repair could decelerate persistent infection progress to IBD.

It has been reported that Lgr5$^{+}$ IESCs at crypt bases are heterogeneous and represent Lgr5$^{hi}$Ki67$^{+}$, Lgr5$^{hi}$Ki67$^{-}$, Lgr5$^{low}$Ki67$^{+}$, and Lgr5$^{low}$Ki67$^{-}$ IESCs, and Lgr5$^{low}$Ki67$^{+}$ IESCs are located above Paneth cells and have significantly higher *Stat5a* mRNA (Basak et al, 2014). These cells can be reactivated toward secretory fates and are also required for regenerating crypts in the presence of severe intestinal damage (Metcalfe et al, 2014), although the required conditions have yet to be established (Basak et al, 2016). Our data indicated that pYSTAT5 induced Lgr5$^{low}$Ki67$^{+}$ IESCs to give rise to a distinct population of ileal crypt cells, the pYSTAT5$^{+}$Lgr5$^{-}$CD24$^{+}$ cells. In this study, we tested the functions and molecular markers of Lgr5$^{low}$Ki67$^{+}$ IESC activation and whether their differentiated progeny play a role in IESC niche regeneration. Our data indicated that pYSTAT5 induced Lgr5$^{low}$Ki67$^{+}$ IESCs to give rise to a distinct population of ileal crypt cells, the pYSTAT5$^{+}$Lgr5$^{-}$CD24$^{+}$ cells. We concluded that a lack of pYSTAT5 impairs the activation of Lgr5$^{low}$Ki67$^{+}$ IESCs and reduces IESC niche formation, leading to susceptibility to *C. difficile* toxins and *C. difficile* infection recurrence, perhaps increasing the comorbidity with ileocolitis. In contrast, pYSTAT5$^{+}$Lgr5$^{-}$CD24$^{+}$ maintains IESCs as niche cells and Ca-pYSTAT5 up-regulates the protective functions of

the Paneth cell niche through feed-forward amplification of Lgr5$^{hi}$ →to Lgr5$^{low}$Ki67$^+$IESCs →to pYSTAT5$^+$Lgr5$^-$CD24$^+$ niche cells (Fig 7F). However, our current studies have some limitations: for example, the unclear genetic or epigenetic mechanisms that a persistent pYSTAT5 regulates Wnt/β-Catenin signaling to restrict Lgr5$^{Low}$Ki67$^+$ cells to differentiate into Paneth niche cells. Interestingly, computational analysis displays some STAT5 GAS motifs on β-Catenin. In the next step of our investigation, besides determining the target genes of pYSTAT5, we will identify the formation of STAT5 tetramer and chromatin remodeling induced by Ca-pYSTAT5, and if the STAT5 tetramers can stabilize intestinal secretory cell differentiation by epigenetically activating or repressing the target genes (Wingelhofer et al, 2018).

Adult IESCs are roughly categorized as either active or quiescent IESCs (Barker et al, 2007; Sangiorgi & Capecchi, 2008). Interestingly, quiescent IESCs are also labeled as a Lgr5$^{low}$Ki67$^+$ or Bmi1$^+$ population that can be reactivated to differentiate hormone-producing enteroendocrine cells or secretory niche cells (Basak et al, 2014, 2016; Yan et al, 2017). Although whether a dedicated pool of quiescent IESCs exists remains unknown, the two populations of IESCs are believed to be regulated by "naive niche" signaling to dynamically switch, permitting IESCs to continually replenish rapidly proliferating progenitors and various types of IECs to maintain intestinal homeostasis (Barker, 2014). Intestinal inflammatory cells secret a special array of regenerative factors that also lead to an injury-induced niche (Li & Clevers, 2010): however, the details of these events have not been thoroughly explored. In our previous study, STAT5 signaling is required for IESC differentiation into mature Paneth cells during injury, reflecting the effects of icS5 on IEC plasticity that the immature Paneth cells might be induced to convert into intermediate progenitors or IESCs (Gilbert et al, 2015; Yu et al, 2018). Nevertheless, niche signaling is essential for IESC plasticity to control intestinal homeostasis and is critically involved in diseases, such as infection, inflammation, and tumorigenesis that occur with severely impaired IEC homeostasis (Nakanishi et al, 2016).

The niche cells are required to provide a microenvironment necessary for the maintenance of IESC self-renewal and proliferation, and thus, these cells are in direct contact with intercalated Lgr5$^+$ IESCs. Depending on the intestinal fragments and injury conditions, niche cells may differ (Sato et al, 2011; Tan & Barker, 2014). Paneth cells secret Wnt ligands (Wnt3a), EGF, and Notch ligands 1 or 4 (Dll1 or 4) to sustain self-renewal of small intestinal Lgr5$^+$ IESCs as well as sense nutrient availability and pathogen invasion by fine-tuning the size of the IESC niche in the small intestine (Tan & Barker, 2014). Therefore, Paneth cells create specialized crypts as an immune environment that restrains pathogens as well as inflammatory toxins and cytokines from targeting IESCs. However, the underlying mechanisms by which niche interaction with IESCs sustains the regeneration of lineages and controls intestinal inflammation or niche differentiation is induced by injuries are unknown (Ge & Fuchs, 2018). Our data indicated that persistent pYSTAT5 activation can restrict IESC differentiation to Paneth cell niche, suggesting that a cytokine-context dependency is a crucial determinant in IESC plasticity and niche regeneration. Thus, our further work will be targeted to identify upstream mediators.

Ectopic Paneth cells can be triggered by various physiological stimuli, such as wounding, cancer, or forced activation of stemness factors (van Es et al, 2005). STAT5 is robustly expressed in the intestinal crypt base and TA zone, and furthermore, STAT5$^+$ IECs show strong co-localization with Lgr5$^+$ intestinal CBCs and crypt niche cells (Gilbert et al, 2015). In this study, we found that persistent icS5 led to Lgr5$^{low}$Ki67$^+$ IESCs towards crypt Paneth cell differentiation, crypt fission, and resistance to various injuries. These data strongly suggest an enhanced crypt immunity that maintains IESC clonogenic activity in the icS5 mice. Consistently, the lineage tracing and kidney transplantation experiments showed that pYSTAT5 enhanced Lgr5 IESC activation towards crypt Paneth cell differentiation. Importantly, in vitro niche replacement experiments showed that secretion by icS5, a persistent pYSTAT5, sustained single IESCs as niche to grow more rapidly than conditioned stem cell media. Together, these results indicate that STAT5 maintains the IESC niche and that loss of STAT5 leads to reduced niche activity. Thus, producing the niche factors induced by Ca-pYSTAT5 could provide a novel avenue to treat persistent enteric infection or gut inflammation and reduce antibiotic resistance. Our future studies will test the direct effects of icS5 on Paneth cell function by Paneth cell or DCS-specific Cre recombinase (Zhao et al, 2018).

Collectively, our results have considerable impact on the field as we have defined critical molecular mechanisms by which IESCs are sustained to regenerate in response to acute intestinal damage, thereby informing novel therapeutic approaches for *C. difficile* infection comorbidity with ileal Crohn's disease, an increasingly common public health problem.

# Materials and Methods

### Materials

All chemicals were purchased from Sigma-Aldrich unless otherwise noted. Antibodies specific for tyrosine phosphorylation–specific STAT5 antibodies (pYSTAT5, Rabbit mAb #9314), cKit (#3074), and Phospho-cKit (pckit, #3391) were purchased from Cell Signaling Technology, antibodies specific for Lgr5 from OriGene (TA503316), antibodies specific for Bmi1 (MAB33341) and ASCL2 (AF6539) from R&D, antibodies specific for Lysozyme (A0099) and Ki67 (M7249) from Dako, antibodies specific for GFP (ab13970), Dclk1 (ab31704), and another Lgr5 antibodies (ab75850) from Abcam, antibodies specific for STAT5a (71-2400) and STAT5b (71-2500) from Zymed Laboratories (Life Technologies), and antibodies specific for STAT5 and β-tubulin from Santa Cruz Biotechnology (sc-835). Another tyrosine phosphorylation–specific STAT5 antibody (Tyr642/699) was from Upstate Biotechnology (Cat. No. 05-495). Toxin A (#152C) and B (#155A) from *C. difficile* were purchased from List Biological Labs. Antibodies for FACS analysis are listed in Table S1. For iPSC culture and HIO maintenance, Matrigel was purchased from BD Bioscience, and human EGF, Noggin, and R-Spondin were from R&D, advanced DMEM/F12 media were from Life Technologies, and mTeSR1 media were from STEMCELL Technologies. For lentiviral transduction reagents, *pLENTI-PGK-Puro-DEST* (w529-2) plasmids were from Addgene, iProof High-Fidelity DNA Polymerase was from Life Science Research, and *pENTR/D-TOPO* cloning kit from Life Technologies. BrdU in situ detection kit was purchased from BD

Biosciences. Click-iT EdU Alexa Fluor 647 FACS and Imaging kits were purchased from Life Technologies. For adult IESC-derived enteroid culture, Matrigel (Cat. No. 356237) was purchased from BD Bioscience, and murine Wnt3a (Cat. No. 1324-WN), EGF (Cat. No. 2028-EG-200), Noggin (Cat. No. 1967-NG-025/CF), and R-spondin (Cat. No. 3474-RS-050) were from R&D Systems. Y27632 was from Stem Cell Technology (Cat. No. 129830-38). Advanced DMEM/F12 media (Cat. No. 12634-010) was from Life Technologies. TRIzol, Applied Biosystems SYBR Green master mixes, and TrypLE cell dissociation reagents were from Thermo Fisher Scientific.

## Animal resources and maintenance

The animal study protocols have been approved by the Institutional Animal Care and Use Committee (IACUC) of Children's Hospital Research Foundation (CHRF), Cincinnati, USA, and Institute of Laboratory Animal Sciences (ILAS), Beijing, P.R. China. *Villin-Cre*ER$^{T2}$ (*VilCre*ER), icS5, or Stat5$^{f/f}$ mice maintained and bred in PI laboratories at Cincinnati and in Beijing (Gilbert et al, 2012a, 2015). *Rosa26-Cre*ER$^{T2}$ (*Rs26Cre*ER), *Rosa26*-tdTomato-*Cre*ER$^{T2}$ (*Rs26*-tdTomato), Lgr5 reporter mice (*Lgr5*-EGFP-IRES-*Cre*ER$^{T2}$, *Lgr5Cre*ER), *Rosa26Cre*ER-mT/mG (*Rosa*$^{mT/mG}$, 007576), *Villin-Cre* (*VilCre*), and C57BL/6 transgenic mice were initially purchased from Jackson Labs. *Rosa26*-LacZ-*Cre*ER$^{T2}$ (*Rs26*-LacZ, N5-1819) were purchased from (Shanghai Model Organism, Shanghai). Constitutive or inducible depletion of STAT5 in IECs was achieved by breeding *Stat5*$^{f/f}$ mice with *VilCre* or *VilCre*ER transgenic mice (el Marjou et al, 2004). *Lgr5Cre*ER and *Rs26*-tdTomato mice were crossed for tracing the progenies of Lgr5$^+$ IESCs (Lgr5-tdTomato), and then Lgr5-tdTomato mice were crossed with *Stat5*$^{f/f}$ or icS5 mice (Lgr5-tdTomato;*Stat5*$^{f/f}$ or icS5) to determine the effect of inducible hyperactivation of icS5 or depletion of *Stat5* on Lgr5 IESC progenies (Hameyer et al, 2007). *Rosa26*$^{mT/mG}$ mice were crossed with *Lgr5Cre*ER;icS5 mice to determine the effects of icS5 upon Lgr5 IESC differentiation in organoids. The effects of inducible depletion of STAT5 upon Lgr5$^+$ IESCs were investigated through crossing *Stat5*$^{f/f}$ mice with *Lgr5Cre*ER and *VilCre*ER mice, by which Tam induction can efficiently deplete STAT5 in Lgr5$^+$ IESCs (Kim et al, 2012). All mice used in these studies have been backcrossed with C57BL/6 for more than 10 generations and were re-genotyped with respect to floxed and Cre *gene*s prior to necropsy. All studies were performed with littermate *Stat5*$^{f/f}$ or icS5 transgenic mice designated as WT controls, *Stat5*$^{f/f}$ mice with *VilCre* or *VilCre*ER denoted as *VilCre;Stat5*$^{f/f}$ or *VilCre*ER;*Stat5*$^{f/f}$ mice, and icS5 mice with *Rs26cre*ER or *VilCre*ER mice denoted as *Rs26Cre*ER; icS5 or *VilCre*ER;icS5 mice. All mice were maintained in specific-pathogen-free conditions in the CHRF and ILAS Animal Care Facility. The animal study protocols were approved by the IACUC at CCHMC, Cincinnati, USA (IACUC2016-0100) and at ILAS, Beijing, P.R. China (HXN16001). Detailed procedures and genotyping for mouse crossing are described in Table S2 and Figs S2C, S3A, and S9A–C.

## *C. difficile* ileitis and colitis models

Male or female mice aged 6–10 wk were treated with a cocktail of antibiotics (0.4 mg/ml kanamycin, 0.035 mg/ml gentamicin, 850 U/ml colistin, 0.215 mg/ml metronidazole, and 0.045 mg/ml vancomycin) in drinking water for 5 d. They were then removed from medicated water for 2 d, at which time they received a single IP injection of clindamycin (30 mg/kg). The purpose of the antibiotic mixture in drinking water followed by the intraperitoneal injection (IP) of clindamycin was to alter the intestinal bacterial composition ("microbiome"). 2 d later, these mice were infected by oral gavage with 1–2.5 × 10$^3$ or 1 × 10$^4$ colony-forming units (CFU) of *C. difficile* spores diluted into 500 µl of DMEM (strain NAP1, ATCC). From that time point forward, the mice were infectious and were housed in a BL2-compatible housing facility at CCHMC. From the time of infection, mice were examined and weighed twice daily for clinical symptoms and weight change. Mice were monitored twice daily during the acute stages of infection (until 96 h after inoculation). *C. difficile* infection is an acute disease, showing maximal symptoms and weight loss at 48–72 h after infection. We found that mice typically became symptomatic at ~36 h and either succumbed or began to recover at 72–96 h after inoculation. Surviving mice were observed and weighed twice daily throughout the experiment and were either euthanized at predetermined time points (days 1–8) or were allowed to recover fully and were euthanized at days 10–21 after infection. At predefined experimental time points after infection, mice were euthanized by carbon dioxide inhalation (2 min) followed by cervical dislocation. The intestines, spleen, and mesenteric lymph nodes were then removed, rinsed in sterile PBS, then either were snap frozen in liquid nitrogen, fixed in PFA, or were placed in cold PBS. These samples were then taken back to our laboratory for RNA extraction, histopathology, or processing for flow cytometry, respectively. The analysis of the ileal and colonic disease severity is as follows: Lesion severity was graded with a system on a scale system as intestinal epithelial hyperplasia and damage (0–3), mucosal neutrophil margination and tissue infiltration (0–3), hemorrhagic congestion and edema of the mucosa (0–3), mucosal necrosis (0–3) (Chen et al, 2008; Hing et al, 2013). These procedures were approved to use in laboratories of Xiaonan Han (PI) IACUC (1E03030) and IBC (2012-0066) at CCHMC, Cincinnati and Xiaonan Han (PI) IACUC (HXN16001) at ILAS, Chinese Academy Institute of Medical Sciences (CAMS) & Peking Union Medical College (PUMC), Beijing.

## Chronic DSS colitis model

Mice were treated orally with three cycles of 7-d 3% DSS with an interval of 5-d water recovery between each cycle of DSS (Fig S1C) (Wirtz et al, 2017). Mucosal histology was evaluated as a total score including intestinal epithelial damage (0–3), ulceration (0–3), and transmural lesion (0–3).

## Lineage-tracing in IECs, in enteroids, or in transplanted HIOs

Lgr5-tdTomato mice were crossed with *icS5* floxed mice (hereafter called Lgr5-tdTomato;Stat5 or icS5), while Lgr5-LacZ mice were crossed with *Stat5* floxed mice (hereafter called Lgr5-LacZ;Stat5) (Barker et al, 2007; Muzumdar et al, 2007; Barker & Clevers, 2010). These compound mice allowed us to trace the effects of gain or loss of function of STAT5 in Lgr5 IESCs upon their progeny following Tam induction. After 1, 7, 15, 30, and 60 d following a single dose of Tam (1 mg/kg), ileums were isolated, immediately flushed with ice-cold PFA (4% formaldehyde, 0.2% glutaraldehyde, and 0.02% NP40 in PBS [which is deficient in Mg$^{2+}$ and Ca$^{2+}$]) and incubated for overnight in a 20-fold volume of the same ice-cold PFA at 4°C on a rolling

platform. The PFA was removed, and the fragments of jejunum, ileums, and colon were washed twice with cold PBS at room temperature on a rolling platform. Jejunal, ileal, or colonic tissues were rolled as "Swiss roll" and embedded with optimal cooling temperature compound (OCT) or paraffin. Tissue sections (4 μM) were prepared and counterstained with DAPI. Red tdTomato ribbons and green GFP Lgr5 progeny were visualized under fluorescence microscope of Leica Live Cell. Lgr5 IESC lineages in the red ribbon were also determined with immunofluorescent staining. The numbers of ileal Paneth cells or colonic Paneth-like cells labeled by tdTomato were counted 14 or 60 d after a single dose of Tam. The PFA-fixed jejunal, ileal, or colonic tissues from Lgr5-RsLacZ;Stat5 mice were under β-galactosidase (LacZ) according to a standard protocol. Tissue sections (4 μM) were prepared and counterstained with HE. Blue LacZ ribbons were imaged under a Leica Live Cell (Fig S3).

*Lgr5CreER-icS5* mice were crossed with *Rosa^{mT/mG}* mice. Lgr5-*Rosa^{mT/mG}*;icS5 mice were generated. Ileal enteroids were differentiated for 14 d and maintained in culture medium with 200 mM 4HT for 4 d. Red tdTomato color was imaged under Leica Live Cell inverted microscope prior to Cre-mediated recombination, whereas green GFP was recorded as video by Leica Live Cell 4 d after 4HT induction (Fig 3D). Lgr5 IESC lineages were determined with immunofluorescent staining (Fig 3E).

LGR5:eGFP BAC reporter hESCs were transduced with lentiviral STAT5a-ER and icS5-ER mutants. HIOs were matured in vitro for 35 d and then were transplanted beneath the kidney capsule of immune-deficient NSG mice. After 2-mo engraftment, the transplanted mice were administered Tam at 25 mg/kg for 3 or 7 or 14 consecutive days. LGR5 lineages in the transplanted HIOs were traced with GFP fluorescent staining (Fig 6G).

### Ileal and colonic crypt isolation, enteroid and colonoids differentiation, and treatments with irradiation, inflammatory cytokines, or bacterial toxins

Ileal or colonic crypts were isolated from *Lgr5Cre*ER;*VilCre*ER;*Stat5^{f/f}* or icS5 and Lgr5-*Rosa26^{mT/mG}*;icS5 mice by manually shaking, then were dissociated with chelation buffer (1 mM EDTA, 5 mM EGTA, 0.5 mM DTT, 43.3 mM sucrose, and 54.9 mM sorbitol) (Sato et al, 2009; Gilbert et al, 2015). After filtered by 70-μm cell strainer, over 200 ileal crypts were re-suspended in Matrigel with 50 ng/ml EGF, 100 ng/ml noggin, and 500 ng/ml R-spondin, or over 200 colonic crypts were re-suspended in Matrigel with EGF, noggin, and R-spondin, 100 ng/ml Wnt3a, and 10 μM Y27432. Enteroids or colonoids were differentiated in the four-well culture plates from days 1 to 14 as published and then frozen in liquid nitrogen (Gilbert et al, 2015). Before any experiments, enteroids or colonoids were recovered and passaged at least two generations. 4HT (200 nM) was used to induce icS5 activation in enteroids after 4-d maturation. Enteroids or colonoids were then exposed to 4 Gy radiation for 10 min on day 4 (Fig 4D and G), or TNFα, toxin A, or toxin B for 4 d on day 4 in the presence or absence of 4HT, to study the effects of icS5 upon irradiation injury, inflammatory cytokines, or bacterial toxins (Fig 5A). Lgr5 IESC self-renewal or proliferation was imaged and quantified as the number of buds (org^{3+}, org^2, org^1, and Sphere) and size of spheres. Viability of enteroids or colonoids was determined by Methylene Blue staining (Figs 5C and S5C). Enteroid or colonoid growth was video-recorded with a Live Image System

(Leica Live Cell, Leica). Lgr5-GFP buds were counted in each well, and the number of buds produced by an individual enteroid or coronoids per day was expressed as budding curves. Organoid-forming capacity in the presence or absence of TNF-α was determined by counting final organoid survival and initial-grown organoids to calculate the percentage of regenerated organoids. Six-replicate wells for each mouse were assessed for each experiment (Fig S5C) (Nakanishi et al, 2016; Schewe et al, 2016).

### FACS

The isolated crypts were dissociated to single-cell or doublet-cell with TrypLE express including 2,000 U/ml DNase (Sigma-Aldrich) for 30 min at 37°C while the enteroids were dissociated with TrypLE express for 15 min at 37°C (Yan et al, 2012; Gilbert et al, 2015; Sasaki et al, 2016). Dissociated IECs were passed through 20-μm cell strainer (Celltrix) and washed with ice cold PBS. Viable IEC single-cells or doublets were gated by forward scatter, side scatter, and pulse–width parameter, and negative staining for 7-ADD (eBioscience) Fig S5A).

FACS analyses of Lgr5, pYSTAT5, Annexin V, CD24, and Ki67 and BrdU were performed by using isolated ileal crypt IECs or dissociated enteroids. IECs were extracted with 2 mM EDTA with manual shaking from crypts or were dissociated from enteroids with TrypLE express, followed by cell strainer filter to generate a single-cell or doublet suspension (Fig S5B). Singlet discrimination was sequentially performed using plots for forward scatter (FSC) (FSC-A versus FSC-H) and side scatter (SSC) (SSC-W versus SSC-H). Dead cells were excluded by scatter characteristics and 7-AAD staining. Lgr5^+ IESCs were identified by their endogenous GFP expression. Populations with Lgr5 and CD24 co-staining were used to determine Lgr5:CD24 doublets along with immunofluorescence (IF). All FACS experiments were performed on an LSR II flow cytometer (BD Biosciences) at the CCHMC or BD FACSAria III cell (BD Bioscience) in the ILAS FACS Facility, and FACS data were analyzed using FlowJo software (Tree Star) (Gilbert et al, 2015).

### Single Lrg5^{hi} intestinal stem cell sorting and "medium replacement" culture

Lgr5 GFP^{hi}, GFP^{low}, and GFP^− IESCs were sorted by BD FACSAria III cell. Single viable Lgr5 cells were gated by forward scatter, side scatter, and pulse–width parameter, and by negative staining for 7-AAD. The sorted cells were manually inspected by inverted microscopy (Fig S5B) (Sato et al, 2009; Yin et al, 2014). Up to 2,000 sorted ileal Lgr5GFP^{hi} IESCs were collected, pelleted, and embedded in Matrigel. Culture medium (Advanced DMEM/F12 supplemented with penicillin/streptomycin, 10 mM Hepes, Glutamax, 1× N2, 1× B27 [all from Invitrogen] and 1 μM N-acetylcysteine [Sigma-Aldrich] containing growth factors 50 ng/ml EGF, 100 ng/ml Noggin, 1 μg/ml R-spondin, and 100 mg/ml Wnt3a) was overlaid. Y-27632 (10 μM) was added for the first 2 d to avoid anoikis. Growth factors and entire medium were replaced every 4 d. Meanwhile, intact ileal crypts were dissociated with EDTA from the ileum of *VilCre*ER or *Rs26Cre*ER;icS5 mice, and enetroids were differentiated from days 1 to 7 as above methods. The culture medium was collected every other day and transferred into the wells containing sorted Lgr5GFP^{hi} IESCs. The

numbers of viable enteroids were calculated in triplicate. Budding curves were created to monitor and compare the growth speed of sorted cell-derived enteroids with or without the transferred medium. For sorting/medium transfer culture experiments, at least three independent experiments were performed. For each experiment, crypts/cells were pooled from three ileums.

### Transduction of LGR5:eGFP H9 cells and generation of HIOs

The generation of the LGR5:eGFP bacterial artificial chromosome (BAC) transgenic reporter H9 hESC line has been previously described (Watson et al, 2014). Briefly, the BAC RP11-59F15 was obtained from the Children's Hospital Oakland Research Institute (http://bacpac.chori.org/) and grown in SW10535 cells. A single colony was expanded in LB +cam at 32°C, and recombineering proteins were induced by incubation at 42°C for 20 min. The recombination cassette consisted of eGFP-FRT-PGKgb2-neo/kan-FRT, 50-bp homology region in LGR5, and a 20-bp homology region to peGFP-PGKneo. The homology regions were selected to replace the initiator methionine of LGR5 with that of eGFP followed by a bovine growth hormone polyadenylation signal and FRT-flanked bifunctional kanamycin/G418 resistance cassette. The recombination cassette was electroporated into SW105 cells, and cells were selected on plates with cam and kanamycin (kan; 50 µg/ml). Clones were analyzed for properly targeted LGR5 BAC by PCR and confirmed by sequencing and nucleofected into single-cell suspensions of H9 hESCs using the Amaxa Human Stem Cell Nucleofector Starter Kit. Cells were selected in G418 (200 ng/ml) for 2 wk. G418-resistant cells were maintained in antibiotic indefinitely.

Production of lentiviral GFP, STAT5a-ER, and icS5-ER iPSCs and LGR5:eGFP was described as in the previously published study (Gilbert et al, 2015). Briefly, LGR5:eGFP or H9hESCs were digested to single cells with Accutase detachment solution (Cat. No. 00-4555-56; Thermo Fisher Scientific), washed and replated at $2 \times 10^5$ per well of a four well dish in mTeSR1 containing 10 µM Y-27632 (Cat. No. 129830-38-2; Sigma-Aldrich) and polybrene (8 µg/ml; Cat. No. 28728-55-4; Sigma-Aldrich). Cells were then transduced by overnight incubation with lentivirus particles. Cells were then fed daily with fresh mTeSR1. Starting 3-d post-transduction, puromycin (2 µg/ml) was added to mTeSR1 to select for transduced cells. Upon reaching semi-confluence, cultures were passaged following the exposure to dispase (1 mg/ml) and gentle trituration to break cells into clumps, and replated in mTeSR1 + puromycin (4 µg/ml). Stably transduced hESC lines were then maintained in an undifferentiated state by daily feeding with fresh mTeSR1 + puromycin (4 µg/ml) and were expanded as colonies using standard dispase passaging methods.

For the generation of HIOs, LGR5:eGFP cells stably expressing Tam-inducible STAT5-ER or icS5-ER constructs and controls were transduced with lentivirus expressing eGFP and were differentiated as previously described (McCracken et al, 2011). Briefly, definitive endoderm was generated by exposure to recombinant human activin A for 3 d (100 ng/ml; Cat. No. 338-AC; R&D Systems). Hindgut endoderm was generated by exposure to recombinant FGF4 (500 ng/ml; Cat. No. 7460-F4-025; R&D Systems) and CHIR99021 (3 µM; Cat. No. 252917-06-9; StemCell Technologies) for 4 d. Spontaneously derived spheroids were then embedded in matrigel (Cat. No. 354234; BD Biosciences) and incubated in media containing recombinant hEGF (50 ng/ml),

hNoggin (100 ng/ml), and hRSpondin1 (500 ng/ml), with complete media changes every 3 d. HIOs were passaged and re-embedded in matrigel after ~2 wk in culture. Approximately 28 d after embedding, HIOs were harvested and used for in vitro permeability assay and transplanted into NSG mice as described in Watson et al, (2014).

### HIO microinjection, FITC-dextran (FD) 4 efflux, and epithelial integrity measurement by TEER

Briefly, using a 7 nl/sec size microcapillary, each HIO was micro-injected an ~2.5 µl mixture of 0.25 ng of TNFα or 50 ng of toxin B or 25 ng of toxin A with 250 ng of fluorescein isothiocyanate-dextran (FD, FD4) (Hill et al, 2017). Microinjection of HIOs. HIOs were gently removed from the Matrigel with a large opening-glass Pasteur glass pipette. Thin-wall glass capillaries (1B100F-4; World Precision Instruments) were pulled using a Sutter micropipette puller (P-97), and microinjection (PLI-100; Harvard Apparatus) was carried out in aseptic condition. FD4 was used in all microinjection experiments to aid in visualizing the injections. Once all of the HIOs were injected, the wells were washed twice with DMEM/F12 medium. The injected HIOs were incubated at 37°C in a 5% $CO_2$ humidified incubator. Then, the media were collected at a given time and the barrier integrity was measured by the signal intensity of FD4. Fluorescence was measured using a fluorescence spectrophotometer (Biotek Instruments) at an excitation wavelength of 492 and an emission wavelength of 515 nm. Permeability was expressed as the mucosal-to-serosal clearance of FD4 (Han et al, 2010). 2-mo-old HIOs were excited open, and lumen intima was inverted. Intima was torn, and each HIO was dissected into three pieces around 1–2 mm$^2$ area and mounted in Snapwell (P2307, area = 0.031 cm$^2$) of EasyMount Ussing chamber systems (Physiological Instrument). TEER from individual piece of HIO was recorded 6, 12 and 24 h after microinjection of FD4, and the average of TEER of each HIO was manually calculated to represent individual HIO epithelial integrity.

### HIO kidney capsule transplantation

H9 or LGR5:eGFP BAC reporter hESCs were transduced with lentiviral STAT5a-ER and icS5-ER mutants (Watson et al, 2014). HIOs were matured in vitro for 35 d and then were transplanted beneath the kidney capsule of immune-deficient NSG mice. Briefly, NSG mice were maintained by an antibiotic chow formula including Sulfamethoxazole (275 ppm, Test Diet) and Trimethoprim (1,365 ppm, Test Diet). Food and water were provided before and after surgeries. The matured HIOs were removed from matrigel, washed with cold phosphate-buffered saline (DPBS; Gibco), and embedded into purified type I collagen (rat tail collagen; BD Biosciences) 12 h before surgery to allow for the formation of a solidified gel plug. These plugs were then placed into standard growth media overnight in intestinal growth medium (Advanced DMEM/F-12, B27, 15 mM Hepes, 2 mM l-glutamine, penicillin-streptomycin) supplemented with 100 ng/ml EGF (R&D). HIOs were then transplanted under the kidney capsule. The mice were anesthetized with 2% inhaled isoflurane (Butler Schein), and the left side of the mouse was then prepped in sterile fashion with isopropyl alcohol and povidine-iodine. A small left-posterior subcostal incision was made to expose the kidney. A subcapsular pocket was created, and the

**Life Science Alliance**

collagen-embedded HIO was then placed into the pocket. The kidney was then returned to the peritoneal cavity, and the mice were given an IP flush of Zosyn (100 mg/kg; Pfizer Inc.). The skin was closed in a double layer, and the mice were given a subcutaneous injection with Buprenex (0.05 mg/kg; Midwest Veterinary Supply) (Fig 6D). At 6–8 wk following the engraftment, the mice were then euthanized or subjected to lineage tracing or radiation experiment.

### Radiation-induced injury mouse models

NSG mice with HIOs transplanted were induced by Tam (1 mg/kg) for 5 d and were exposed to 12 Gy whole body γ-radiation for 10 min at Comprehensive Mouse and Cancer Core, CCHMC (Hua et al, 2012). The mice were intraperitoneally administered with BrdU 3.5-d post-radiation and then euthanized 3 h later. The intestinal tissues were inspected for gross and histological abnormalities (Han et al, 2010; Hua et al, 2012). Radiation Injury Scores (RIS) and mucosal ulcer-ation of radiation-induced intestinal mucositis were determined as previously described (Akpolat et al, 2009). RIS is a composite his-topathologic scoring system that has been extensively improved by our laboratory. Briefly, the scores of γ-radiation-induced ileal mucosal damages (scores: 1, 2, and 3), crypt loss (scores: 1, 2, and 3), mucosal ulcerations (scores: 1, 2, and 3), and thickening of intestinal wall (scores: 1, 2, and 3) were combined as RIS. The numbers of proliferating crypts or regenerated crypts ("microcolonies") were quantified as crypts per millimeter under microscope (magnifica-tion 100×) and were confirmed by BrdU labeling (Gilbert et al, 2015).

### RNA-seq

Ileal or colonic crypts were isolated by shaking intestinal tissue in PBS with 10 mM EDTA at 37°C for 30 min (Fig S8F). Total RNA was extracted using mirVana miRNA Isolation kit (Thermo Fisher Sci-entific) with total RNA extraction protocol. The RNA concentration was measured by NanoDrop (Thermo Fisher Scientific), and its integrity was determined by Bioanalyzer (Agilent). The RNA input was 1 μg of RNA from each sample and they were pooled for cDNA library construction and sequencing on an Illumina HiSeq in-strument with 1 × 75 bp reads at the Genomics, Epigenomics and Sequencing Core, Department of Environmental Health, University of Cincinnati. The BaseSpace (Illumina) applications TopHat Align-ment (Version 1.0.0) and Cufflinks Assembly and Differential Ex-pression (Version 1.1.0) were used to produce aligned reads and to perform novel transcript assembly and differential expression analysis. The Ingenuity Pathway Analysis tool from QIAGEN was used for pathway analysis and upstream regulator analysis of significant genes that showed more than twofold of differential expression. Finally, these co-regulated genes were subjected to the Distant Regulatory Elements (DiREs) server to predict common regulatory elements (Gotea & Ovcharenko, 2008).

### Immunoblotting, immunohistochemistry, IF, whole-mount intestinal LacZ staining and quantitated real-time PCR (qRT-PCR)

Isolated ileal IECs, HIOs, and mucosal tissue were saved. Total cellular protein (TP), cytosolic protein, and nuclear protein (NE) extracts were prepared using cold RIPA buffer and the NE-PER kit per the manufacturers' recommendations (Pierce). Expression of Lgr5, Bmi1, Dclk1, Lyso, and intercellular junctional proteins (ZO-1 or 2, Occludin, Claudin 1 and 2, E-Cadherin) in HIOs were measured in TP. The nuclear abundance of β-catenin and cleavage of the Notch intracellular domain protein (NICD), pYSTAT5, and STAT5a were detected in NE (Gilbert et al, 2012). Frozen tissues from mouse ileums and enteroids, and HIOs were prefixed in PFA. Sections (4 μm) were labeled with pYSTAT5, ASCL2, BrdU, Ki67, Lyso, Muc2, E-Cadherin, CHGA, Lyso, ZO-1 or 3, Ki67 antibodies following FITC-conjugated or TRITC-conjugated anti-rabbit secondary antibodies, and DAPI was used for nuclear counterstaining. pYSTAT5 (phosphotyrosine), BrdU, and Ki67 were examined in paraffin-embedded intestinal sections using VECTASTAIN Elite ABC system (Vector lab). The full length of intestinal and colonic fragments were isolated from Tam-treated Lgr5-LacZ;Stat5 mice and immediately fixed with PFA LacZ solution. The fixed intestines were then stained with freshly made LacZ substrate overnight. Whole-mount intestinal analysis of in vivo lineage tracing was performed under 3D microscope (Zeiss). The images were scanned for overall measuring LacZ positive mucosa. Finally, the intestinal fragments were embedded and sectioned to analyze positive LacZ staining crypts under a microscope. Total RNA was extracted from mouse tissues or cultured enteroids using RNeasy Mini kit (QIAGEN) according to the manufacturer's protocol. PCRs were performed with SYBR Green qRT-PCR mix in the Mx3000p thermocycler (Stratagene) (Gilbert et al, 2012). The primer sequences (Zhang et al, 2015b; Gilbert et al, 2015; Nakanishi et al, 2016; Smith et al, 2017) are listed in Table S3.

### Statistics

For all data analysis, the statistics software package SPSS (version 19.0) or GraphPad Prism (7.0) was used. Univariate survival analysis was subsequently carried out separately for each investigated pa-rameter using Kaplan–Meier estimates. Survival curves were com-pared and assessed using the log-rank test. $P$-values of 0.05 or less were considered significant when using $t$ tests and ANOVA. RNA-seq data were analyzed by biostatisticians at University of Cincinnati and CCHMC, Cincinnati, OH, USA, University of Veterinary Medicine Vienna, Vienna, Austria, and ILAS, CAMS, and PUMC, Beijing, P.R. China.

### Transcript profiling

GSE112607.

# Study approval

The animal study protocols were approved by the IACUC at CCHMC, Cincinnati, USA (IACUC2016-0100), and at ILAS, CAMS, and PUMC, Beijing, P. R. China (HXN16001).

# Supplementary Information

# Acknowledgements

This work was supported by Crohn's Colitis Foundation of America Senior Research Award (426234, to X Han), NIAID R21 (AI103388 to X Han) and Cincinnati Children's Hospital Research Foundation Digestive Health Center (P30DK078392), NIH R01 (DK098231) (to L Denson), USA, and Peking Union Medical College Professor Scholar (2016RC310011 to X Han), P.R. China. R Moriggl was supported by a private cancer metabolism grant donation from Liechtenstein and the Austrian Science Fund (FWF), grants SFB F4707 and SFB-F06105, Austria.

## Author Contributions

R Liu: data curation, formal analysis, and methodology.
R Moriggl: conceptualization, resources, formal analysis, investigation, and editing.
D Zhang: data curation and formal analysis.
H Li: data curation, formal analysis, and methodology.
R Karns: software and formal analysis.
H-B Ruan: formal analysis and methodology.
H Niu: data curation and methodology.
C Mayhew: data curation, methodology, and funding acquisition.
C Watson: data curation.
H Bangar: data curation.
S-w Cha: data curation and methodology.
D Haslam: conceptualization, formal analysis, and methodology.
T Zhang: software and formal analysis.
S Gilbert: data curation.
N Li: data curation.
M Helmrath: conceptualization, resources, and methodology.
J Wells: conceptualization, resources, methodology, investigation, and funding acquisition.
L Denson: conceptualization, formal analysis, methodology, and funding acquisition.
X Han: conceptualization, resources, formal analysis, data curation, supervision, funding acquisition, validation, investigation, visualization, methodology, project administration, writing—original draft, review, and editing.

## Conflict of Interest Statement

The authors declare that they have no conflict of interest.

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
