## [Reviewer comments · Life Science Alliance]

Life Science Alliance

Constitutive STAT5 Activation Regulates Paneth and Paneth-like Cells to Control *C. difficile* Colitis

Ruixue Liu, Richard Moriggl, Dongsheng Zhang, Haifeng Li, Rebekah Karns, Hai-Bin Ruan, Haitao Niu, Christopher Mayhew, Carey Watson, Hansraj Bangar, Sang-wook Cha, David Haslam, Tongli Zhang, Shila Gilbert, Na Li, Michael Helmrath, James Wells, Lee Denson, Xiaonan Han
DOI: <https://doi.org/10.26508/lsa.201900296>

Corresponding author(s): Xiaonan Han, Cincinnati Children's Hospital Medical Center

Review Timeline:

Submission Date:	2019-01-17
Editorial Decision:	2019-02-07
Revision Received:	2019-03-23
Editorial Decision:	2019-03-26
Revision Received:	2019-03-27
Accepted:	2019-03-27

Scientific Editor: Andrea Leibfried

Transaction Report:

February 7, 2019

Re: Life Science Alliance manuscript #LSA-2019-00296

Prof. Xiaonan Han
Cincinnati Children's Hospital Medical Center
Gastroenterology, Hepatology and Nutrition
Burnet Ave
3333 Burnet Ave
Cincinnati, OH 45229

Dear Dr. Han,

Thank you for submitting your manuscript entitled "Constitutive STAT5 Activation Regulates Paneth Cells to Control *C. difficile* Ileocolitis" to Life Science Alliance. The manuscript was assessed by expert reviewers, whose comments are appended to this letter.

As you will see, the reviewers appreciate your data and provide constructive input on how to further strengthen the conclusions drawn and the presentation of the data of your manuscript. We would thus like to invite you to submit a revised version of your work, following the suggestions of the reviewers.

Thank you for this interesting contribution to Life Science Alliance. We are looking forward to receiving your revised manuscript.

Sincerely,

- A letter addressing the reviewers' comments point by point.
- An editable version of the final text (.DOC or .DOCX) is needed for copyediting (no PDFs).
- High-resolution figure, supplementary figure and video files uploaded as individual files: See our detailed guidelines for preparing your production-ready images, <http://life-science-alliance.org/authorguide>
- Summary blurb (enter in submission system): A short text summarizing in a single sentence the study (max. 200 characters including spaces). This text is used in conjunction with the titles of papers, hence should be informative and complementary to the title and running title. It should describe the context and significance of the findings for a general readership; it should be written in the present tense and refer to the work in the third person. Author names should not be mentioned.

B. MANUSCRIPT ORGANIZATION AND FORMATTING:

Full guidelines are available on our Instructions for Authors page, <http://life-science-alliance.org/authorguide>

Reviewer #1 (Comments to the Authors (Required)):

This is a very interesting paper investigating the role of STAT5 in IESC homeostasis and the response to *C. difficile* infection. The authors have generated a series of transgenic mice whose utilization in quite elegant tracing experiments, together with *ex vivo* experiments in organoids, definitively established the role of STAT5 loss of function or gain of function in the control of IESC

lineage specification as well as its role under conditions of colitis induced by *C. difficile*. Importantly they establish that the loss of STAT5 results in Paneth cell dysfunction, which makes these mice more vulnerable to the pathological effects of *C. difficile*. They also do the inverse set of experiments using a permanently active mutant of STAT5 that led to very interesting and consistent results. These are well performed studies that are of potential great significance and of relative high novelty.

The last part of the paper is a little bit weaker. Thus, although some experiments were performed to try to explain mechanistically how STAT5 works at a more molecular level, these more mechanistic studies lack the depth and detail that would make of this a really highly impactful paper. In any case, as it is now, the results presented, although somehow correlative at the molecular level, are of great interest and need to be reported because the mouse *in vivo* studies are solid and of interest to the signaling and intestinal research community.

The paper is in need of some English editing to improve the flow and make it easier to follow. There are also some panel labels that seem to have been switched that need to be fixed and in general the presentation of the figures and the text could use some more careful editing.

Reviewer #2 (Comments to the Authors (Required)):

The manuscript by Liu et al. focuses on an important issue for *C. difficile* infection and tissue regeneration: Paneth cell maturation. Overall I find the manuscript suitable for the journal, however there are some points that need to be changed and/or answered.

Major points:

- In the discussion the authors claim "Our data indicated that IESC differentiation to Paneth cell niche requires a persistent pYSTAT5 activation" yet in Figure 2a .the epithelial knockout still shows lysozyme positive Paneth cells.
- Is pYSTAT5 downregulated in chronic colitis models?
- Paneth cell quantification in figure 3 would be clearer if repeated with another methodology (flow cytometry, western blot)
- It would be interesting to see if the different STAT5 mouse models presented have a difference in budding formation capacity, something considered as a good output for regenerative capacity of an organoid in homeostasis and inflammation.

Minor points:

- How does STAT5 activate Wnt?
- The term Paneth-like cell was first coined in this manuscript "Schewe et al. 2016 Cell Stem Cell" as such this reference should be added.

We are appreciative of the insightful comments of Editorial Board Members and the Reviewers. We have uploaded the revised manuscript and provide you with a point-by-point response where we have addressed each of the major and minor concerns. Changes in the text body have been underlined for ease of review. We believe that with additional new data focused on comments we received, the revised manuscript is substantially improved. We hope that our revised manuscript and responses to critiques can be further considered for a publication by the reviewers and the Editorial Board of *Life Science Alliance*. A point-by-point response is included below.

Reviewer #1 (Comments to the Authors (Required)):

1. This is a very interesting paper investigating the role of STAT5 in IESC homeostasis and the response to C. difficile infection. The authors have generated a series of transgenic mice whose utilization in quite elegant tracing experiments, together with ex vivo experiments in organoids, definitively established the role of STAT5 loss of function or gain of function in the control of IESC lineage specification as well as its role under conditions of colitis induced by C. difficile. Importantly they establish that the loss of STAT5 results in Paneth cell dysfunction, which makes these mice more vulnerable to the pathological effects of C. difficile. They also do the inverse set of experiments using a permanently active mutant of STAT5 that led to very interesting and consistent results. These are well performed studies that are of potential great significance and of relative high novelty.

Author response: Thank so much for an overall positive comment on our research significance. I have revised our Summary Blurb as “Constitutively active pYSTAT5 restricts intestinal stem cells to give rise to pYSTAT5⁺ niche cells by regulating Wnt/β-Catenin. Lack of pYSTAT5 decreases Paneth cells to exaggerate *C. difficile* colitis”, where we reiterate our finding. We conclude that STAT5 is an essential protein to specify niche cell differentiation.

2. The last part of the paper is a little bit weaker. Thus, although some experiments were performed to try to explain mechanistically how STAT5 works at a more molecular level, these more mechanistic studies lack the depth and detail that would make of this a really highly impactful paper. In any case, as it is now, the results presented, although somehow correlative at the molecular level, are of great interest and need to be reported because the mouse in vivo studies are solid and of interest to the signaling and intestinal research community.

Author response: Very appreciate your critiques. Our on-going study is focused on “Utilize ChIP-seq assay to determine the target genes by which Ca-pYSTAT5 differentially regulates secretory cell lineage commitment in intestines or colon”, part of which will involve a tetrameric STAT5 augments EZH2 activity, subsequently stimulating epigenetic markers to induce or repress lineage-specific genes. We have included the limitation of our current research, and future research targets in the discussion in **line 13 on Page 23**.

Author response:

3. The paper is in need of some English editing to improve the flow and make it easier to follow. There are also some panel labels that seem to have been switched that need to be fixed and in general the presentation of the figures and the text could use some more careful editing.

Author response: We have broadly revised our manuscript and our data Figures to match Figure legends. We have also included an English native speaker to improve the flow and to make it easier to follow.

Reviewer #2 (Comments to the Authors (Required)):

The manuscript by Liu et al. focuses on an important issue for C. difficile infection and tissue regeneration: Paneth cell maturation. Overall, I find the manuscript suitable for the journal, however there are some points that need to be changed and/or answered.

Author response: We appreciate all comments from **reviewer 2**. We respond to them with new data point by point. The revised parts are underlined.

Major points:

1. *In the discussion the authors claim "Our data indicated that IESC differentiation to Paneth cell niche requires a persistent pYSTAT5 activation" yet in Figure 2A. the epithelial knockout still shows lysozyme positive Paneth cells.*

Author response:

We reported that STAT5 acts as an intrinsic factor to regulate IESC proliferation (Gilbert et al, 2015). In our revised paper, we supplement a 14 day-lineage tracing data using *Lgr5-LacZ;Stat5* mice and a single dose Tam-induction (**Supplementary Fig S4A**). Our new data show that loss of STAT5 in *Lgr5*⁺ IESCs leads to absence or significantly reduced LacZ staining in the jejunal, ileal and colonic crypt cells, indicating that STAT5 signaling regulates IESC stemness as an intrinsic signaling (**Supplementary Fig S4C-D**). The mouse model with a constitutively active pYSTAT5 (Ca-pYSTAT5) showed an increased numbers of Paneth cells at the crypt bases and villi (**Figure 2E**). Thus, STAT5 may regulate Paneth cells through specifying Paneth cell fate or controlling Paneth cell maturation. Based on the lineage tracing data from STAT5 loss function in mice, we focused on if Ca-pYSTAT5 can directly induce IESC differentiation to increase the number of Paneth cells. Our results showed that Ca-pYSTAT5 increased IESC lineage tracing at the crypt bases in the *Lgr5-Rs26tdtomato;icS5* mice (**Figure 3B**, and **Supplementary Fig S3B** and **C**) and led to *de novo* Paneth cell differentiation in the HIO xenotransplantation (**Fig 6H**). Together, these data indicate that Ca-pYSTAT5 in IESCs increase Paneth cell differentiation. Therefore, our experiment provides a new transcription factor that could restrict *Lgr5* IESC differentiation to Paneth cells. Considering that our result displayed the possible binary effects of pYSTAT5 on Paneth cell maturation and the differentiation of *Lgr5*^{low}Ki67⁺ into pYSTAT5⁺*Lgr5*⁻CD24⁺Lyso⁺ niche cells, we toned down our explanation of results and revised it into: "a persistent pYSTAT5 activation can restrict IESC differentiation to Paneth cell niche", in **line 2** on **page 25**. We will cross a *Defensin 6a* promoter-driven Cre transgenic mouse line and *icS5* floxed mice to test the Paneth cell phenotypes affected by Ca-pYSTAT5 in the future. We have included it in the discussion **in line 19** on **page 25**.

Using two markers of mature Paneth cells (Lyso and CD24), we observed that *Stat5*-deficient mice exhibited a significant reduction of Lyso⁺ Paneth cell numbers at the crypt bottom shown in **Figure 2A**, which is consistent with a reduced level of *Lyso* and *Defensin* expression at the *Stat5*-deficient crypts shown in **Figure 2B** and a diminished number of CD24⁺ crypt Paneth cells in the *Stat5*-deficient mice determined by FACS analysis in **Figure 4A**. Knock-out of STAT5 reduced the number of Paneth cells, but neither Lyso⁺ nor CD24⁺ Paneth cells are completely disappearing. STAT5 signaling could cooperate with other pathways to control Paneth cell differentiation. These data suggest that compensatory pathways independent of STAT5 also regulate IESCs towards Paneth cell differentiation. Thus, depletion of STAT5

might may lead to alternative mechanisms or other pathway activation or inhibition, such as interferon (Farin et al, 2014), MAPK (Heuberger et al, 2014) or Notch signaling (Rothenberg et al, 2012), which were reported to change Paneth cell differentiation or function. We have included some discussion in **line 8** on **page 7** To address these potential mechanisms will require more work beyond the scope of this manuscript, we will testify them in the future.

2. Is pYSTAT5 downregulated in chronic colitis models?

Author response: As the reviewer suggested, we have provided a set of new data showing colonic mucosal pYSTAT5 during chronic DSS colitis. We created a chronic DSS colitis model by orally giving WT mice with three cycles of 7-day 3% DSS treatment with an interval of 5-day water recovery between each cycle of DSS (**Supplementary Fig S1C**) (Wirtz et al, 2017). We found an increased pYSTAT5 immune-staining in the nascent crypts pointed by the arrow heads, demonstrating the importance of pYSTAT5 in the colonic IEC regeneration during chronic colitis. We have included these data in the **Supplementary Fig S1C** and result description from **line 14 to 20 on Page 8**.

3. Paneth cell quantification in figure 3 would be clearer if repeated with another methodology (flow cytometry, western blot).

Author response: Given that the amount of protein from a single organoid is too low to be detected by immunoblotting, and the immunostaining is not quantitated, we chose the real-time PCR and found an upregulated *Lyz1* expression (Upper right panel of **Figure 3E**), which is consistent with the result showed in **Figure 1B**. In addition, to demonstrate the upregulated *Lyz1* through increased Paneth cell differentiation, we measured *Sox9* expression with real-time PCR. It shows the upregulated *Sox9* expression in the same icS5-induced enteroids, which is consistent with increase *Sox9* protein staining in the **supplementary Fig S4E**. These new data are respectively included in **Figure 3E** and **Supplementary Fig S4F** and in **line 4 page 12** in the revised manuscript.

4. It would be interesting to see if the different STAT5 mouse models presented have a difference in budding formation capacity, something considered as a good output for regenerative capacity of an organoid in homeostasis and inflammation.

Author response: According to Dr. Fodde group study (Schewe et al, 2016), we have supplemented the data showing organoid forming capacity in the presence or absence of TNF- α by counting final organoid survival and initial grown organoids to calculate the percentage of regenerated organoids. Considering the length of **Figure 5A**, we supplement a graph of organoid forming capacity as a **supplementary Fig S6A and B** to show the organoid forming capacity in the STAT5 Δ IEC+/-, STAT5 $^{+/+}$ and STAT5 Δ IEC+++ mice. We also move the number of Org $^{1-3}$ and the size of spheroid of the controls from the original **Supplementary Figure 6A** to **Figure 3A**, which shows enteroid multiplicity at homeostasis and during inflammation. Accordingly, these data showed that LOF of STAT5 reduced the organoid forming capacity, characterized by the reduced budding numbers and enlarged spheroid size (**supplementary Fig S6B**) and reduced percentage of survival organoid numbers (**supplementary Fig S6B**), the phenotypes of which were significantly exaggerated by the presence of TNF- α . These data confirm our previous reports and extend the role of STAT5 in IESCs during inflammation, illuminating STAT5 engagement in a 3D *in vitro* primary organoid system with different genetic mutation. We have included these results from **line 3 to 8 and line 11 on Page 16**.

Minor points:

1. How does STAT5 activate Wnt?

Author response: In the **Figure 7C** and **Supplementary Fig S7C**, we showed that STAT5 activation (pYSTAT5) resulted in an increased nuclear β -Catenin in the IESCs, demonstrating that pYSTAT5 amplified Wnt/ β -catenin signaling to enhance IESC self-renewal and expand the IESC compartment (**Figure 2E**). We also found a higher level of Sox9 expression and increased Paneth cell numbers found in the enteroids with pYSTAT5 (**Figure 7D** and **Supplementary Fig S4F**). These data suggest the pYSTAT5-induced higher level of Wnt promotes the Lgr5⁺ IESCs towards Paneth cell differentiation. Using TOPflash Wnt reporter-transduced iPSCs or colon cancer cell lines that have a constitutive Wnt activation (Fuerer & Nusse, 2010), we will address in the next step if STAT5 can bind β -Catenin/Tcf4 gene locus to directly activate Wnt signaling. We have included this design in the future study and stated it in the discussion **from line 13 to 20 on page 23**.

Interestingly, we found that a persistent pYSTAT5 (Ca-pYSTAT5) induced by icS5-ER led a lower level of nuclear β -Catenin while higher level of Lyso in the iPSC-derived HIOs compare to their expression in the transient pYSTAT5 induced by STAT5A-ER (**Supplementary Fig S7C**). Since Ca-pYSTAT5 has chromatin remodeling power and computation analysis shows there are many TCCN₍₂₋₅₎GAA GAS motifs on the *β -Catenin* (Wingelhofer et al, 2018), we postulate that Ca-pYSTAT5 epigenetically controls Wnt target gene expression. For instance, Ca-pYSTAT5 may induce a formation of STAT5 tetramers on *β -Catenin* gene, which may lead to chromatin remodeling and epigenetically induce the repression of Wnt target genes controlling stemness while the activation of genes inducing Paneth cell differentiation. We will use CHIP-seq, ATAC-seq and RNA-seq analyses to determine the possible mechanisms by which tetrameric STAT5 stimulates epigenetic markers to induce or repress lineage-specific genes. However, it is beyond the scope of this manuscript. We have included it in the future direction **from line 13 to 20 on page 23**.

2. The term **Paneth-like cell** was first coined in this manuscript "Schewe et al. 2016 Cell Stem Cell" as such this reference should be added.

Author response: We have cited this reference in **line 11** on **page 5** and have included them as **Supplementary Methods**.

References

Farin HF, Karthaus WR, Kujala P, Rakhshandehroo M, Schwank G, Vries RG, Kalkhoven E, Nieuwenhuis EE, Clevers H (2014) Paneth cell extrusion and release of antimicrobial products is directly controlled by immune cell-derived IFN- γ . *J Exp Med* 211: 1393-1405

Fuerer C, Nusse R (2010) Lentiviral vectors to probe and manipulate the Wnt signaling pathway. *PLoS One* 5: e9370

Gilbert S, Nivarthi H, Mayhew CN, Lo YH, Noah TK, Vallance J, Rulicke T, Muller M, Jegga AG, Tang W et al (2015) Activated STAT5 Confers Resistance to Intestinal Injury by Increasing Intestinal Stem Cell Proliferation and Regeneration. *Stem Cell Reports* 4: 209-225

Heuberger J, Kosel F, Qi J, Grossmann KS, Rajewsky K, Birchmeier W (2014) Shp2/MAPK signaling controls goblet/paneth cell fate decisions in the intestine. *Proc Natl Acad Sci U S A* 111: 3472-3477

Rothenberg ME, Nusse Y, Kalisky T, Lee JJ, Dalerba P, Scheeren F, Lobo N, Kulkarni S, Sim S, Qian D et al (2012) Identification of a cKit(+) colonic crypt base secretory cell that supports Lgr5(+) stem cells in mice. *Gastroenterology* 142: 1195-1205 e1196

Schewe M, Franken PF, Sacchetti A, Schmitt M, Joosten R, Bottcher R, van Royen ME, Jeammet L, Payre C, Scott PM et al (2016) Secreted Phospholipases A2 Are Intestinal Stem Cell Niche Factors with Distinct Roles in Homeostasis, Inflammation, and Cancer. *Cell Stem Cell* 19: 38-51

Wingelhofer B, Neubauer HA, Valent P, Han X, Constantinescu SN, Gunning PT, Muller M, Moriggl R (2018) Implications of STAT3 and STAT5 signaling on gene regulation and chromatin remodeling in hematopoietic cancer. *Leukemia* 32: 1713-1726

Wirtz S, Popp V, Kindermann M, Gerlach K, Weigmann B, Fichtner-Feigl S, Neurath MF (2017) Chemically induced mouse models of acute and chronic intestinal inflammation. *Nat Protoc* 12: 1295-1309

March 26, 2019

RE: Life Science Alliance Manuscript #LSA-2019-00296R

Prof. Xiaonan Han
Cincinnati Children's Hospital Medical Center
Gastroenterology, Hepatology and Nutrition
Burnet Ave
3333 Burnet Ave
Cincinnati, OH 45229

Dear Dr. Han,

Thank you for submitting your revised manuscript entitled "Constitutive STAT5 Activation Regulates Paneth and Paneth-like Cells to Control *C. difficile* Colitis". As you will see, reviewer #2 appreciates the introduced changes and we would thus be happy to publish your paper in Life Science Alliance pending final revisions necessary to meet our formatting guidelines:

- please upload all figures (also S figures) as individual files
- it would be good in my view to incorporate the suppl material and methods into the main manuscript file to allow for easier accessibility
- please note that not all authors are listed in our submission system and that author Ruixue Liu is spelled differently - please add/fix.
- you mention figure 8C in the ms text, but this figure doesn't exist - please check and fix
- please mention figure S8F and the tables in the manuscript text (currently missing)
- please check figures 2-7, S1-S6 and S9 and add scale bars where missing
- please add the statistical test used into each figure legend (t-test / ANOVA)
- please note that we adhere to ICMJE authorship/author contribution guidelines; please check each author's contribution again

A. FINAL FILES:

B. MANUSCRIPT ORGANIZATION AND FORMATTING:

Sincerely,

Andrea Leibfried, PhD
Executive Editor
Life Science Alliance
Meyerhofstr. 1
69117 Heidelberg, Germany
t +49 6221 8891 502

e.a.leibfried@life-science-alliance.org
www.life-science-alliance.org

Reviewer #2 (Comments to the Authors (Required)):

I am very pleased with how the authors answered the points and want to congratulate on an interesting manuscript that will improve the knowledge of the literature in this field. I have no further comments and support this manuscript for publication.

March 27, 2019

RE: Life Science Alliance Manuscript #LSA-2019-00296RRR

Prof. Xiaonan Han
Cincinnati Children's Hospital Medical Center
Gastroenterology, Hepatology and Nutrition
Burnet Ave
3333 Burnet Ave
Cincinnati, OH 45229

Dear Dr. Han,

Thank you for submitting your Research Article entitled "Constitutive STAT5 Activation Regulates Paneth and Paneth-like Cells to Control *C. difficile* Colitis". It is a pleasure to let you know that your manuscript is now accepted for publication in Life Science Alliance. Congratulations on this interesting work.

DISTRIBUTION OF MATERIALS:

Again, congratulations on a very nice paper. I hope you found the review process to be constructive and are pleased with how the manuscript was handled editorially. We look forward to future exciting submissions from your lab.

Sincerely,

Andrea Leibfried, PhD
Executive Editor
Life Science Alliance
Meyerohofstr. 1
69117 Heidelberg, Germany
t +49 6221 8891 502
e a.leibfried@life-science-alliance.org
www.life-science-alliance.org